# The NELF pausing checkpoint mediates the functional divergence of Cdk9

Michael DeBerardine [1], Gregory T. Booth[1,2], Philip P. Versluis[1] & John T. Lis [1] ✉

Promoter-proximal pausing by RNA Pol II is a rate-determining step in gene transcription that is hypothesized to be a prominent point at which regulatory factors act. The pausing factor NELF is known to induce and stabilize pausing, but not all kinds of pausing are NELF-mediated. Here, we find that NELF-depleted *Drosophila melanogaster* cells functionally recapitulate the NELF-independent pausing we previously observed in fission yeast (which lack NELF). Critically, only NELF-mediated pausing establishes a strict requirement for Cdk9 kinase activity for the release of paused Pol II into productive elongation. Upon inhibition of Cdk9, cells with NELF efficiently shutdown gene transcription, while in NELF-depleted cells, defective, non-productive transcription continues unabated. By introducing a strict checkpoint for Cdk9, the evolution of NELF was likely critical to enable increased regulation of Cdk9 in higher eukaryotes, as Cdk9 availability can be restricted to limit gene transcription without inducing wasteful, non-productive transcription.

For most genes in higher eukaryotes, RNA Polymerase II (Pol II) enters into a stable, paused state before proceeding into productive gene transcription. This *promoter-proximal pausing* event has become increasingly recognized as a prominent rate-limiting step in gene transcription, and thus a likely stage at which regulatory processes can act to modulate gene expression[1,2].

Two pausing factor complexes, NELF and DSIF, are known to bind Pol II and induce a stably paused state[3–5], but DSIF remains bound after pause-release and promotes elongation as well[1]. NELF's role as a pausing factor is well supported by functional studies in vitro and in vivo, but recent evidence indicates that not all promoter-proximal pausing is dependent on NELF[6–8]. Still, NELF substantially increases in vivo pausing half-lives, and pausing is sharply reduced following its depletion[9–15].

In higher eukaryotes, the release of paused Pol II into productive elongation is acutely dependent on the kinase activity of the Cdk9 subunit of the P-TEFb complex[1,16–18]. While the immediate functional target(s) of Cdk9 that mediates pause-release remains a subject of ongoing research[19], Cdk9 kinase activity is necessary in vitro to suppress formation of a NELF-bound paused complex in favor of an active complex bound by elongation factors[20]. Such a coordinated

exchange of NELF for elongation factors may be a critical aspect of the pause-release mechanism.

However, Cdk9's essential role in promoting transcription elongation exists independently of NELF. Cdk9 homologs predate the evolution of NELF and are known to be essential transcriptional kinases in NELF-lacking eukaryotes[21–24]. We previously found that the fission yeast *Schizosaccharomyces pombe* (which lacks NELF) has modest pause-like accumulations of Pol II at the 5′ ends of genes[7]. However, unlike in higher eukaryotes, inhibition of Cdk9 in *S. pombe* does not prevent gene body entry by Pol II, but Pol II complexes that enter the gene body in the absence of Cdk9 activity have reduced elongation rates[25].

Here, we find that only NELF-mediated pausing enforces a strict early checkpoint for Cdk9, which efficiently shuts down gene transcription following loss of Cdk9. Without NELF, *Drosophila* cells have the same phenotypic response to Cdk9 inhibition as *S. pombe*, which is characterized by global perturbations to elongation that render Pol II unable to transcribe beyond a few kilobases. As NELF allows cells to efficiently respond to a restriction in Cdk9 activity, we propose that NELF is critical for enhancing the regulatory role for Cdk9 in higher eukaryotes, as modulating the availability of Cdk9 becomes an

---

[1]Department of Molecular Biology and Genetics, Cornell University, Ithaca, NY, USA. [2]Present address: Kanvas Biosciences, Monmouth Junction, NJ, USA. ✉e-mail: johnlis@cornell.edu

efficient means of modulating gene transcription in NELF-containing organisms.

## Results

### Pausing is substantially reduced at nearly all Pol II promoters following NELF depletion

To investigate the mechanistic relationship between NELF and Cdk9 in vivo, we used RNAi to deplete NELF from *Drosophila melanogaster* S2 cells, which we then treated with the specific Cdk9 inhibitor Flavopiridol (FP) (Fig. 1a). We targeted the NELF-E subunit as its depletion is known to destabilize other subunits[15]. In our hands, this depletion is effective, but incomplete, at the mRNA and protein levels (Supplementary Fig. 1a, b, Supplementary Fig. 7a).

To measure the immediate effect of Cdk9 inhibition on active transcription in NELF-depleted cells, we used PRO-seq, which measures engaged RNA polymerases genome-wide at single-base resolution with high specificity and sensitivity[26–28]. We also prepared polyA+ RNA-seq libraries from the same cellular material to measure changes in stable mRNA[29]. To account for global changes in active transcription, we used exogenous spike-in cells (mouse embryonic fibroblasts) to normalize our PRO-seq and mRNA-seq libraries across different conditions. Our PRO-seq and mRNA-seq libraries are reproducible, both in terms of the spike-in quantifications as well as the genomic read distributions (Supplementary Fig. 1c–f).

Spike-in quantifications of our PRO-seq libraries show that NELF depletion and Cdk9 inhibition both significantly decrease the number of engaged RNA polymerases detected on a per-cell basis (Fig. 1c, Supplementary Fig. 1e). NELF-depleted cells have substantially reduced Pol II density across nearly all promoter-proximal regions, including both highly and lowly paused genes (to the extent that changes can be reasonably measured) (Fig. 1b left, Supplementary Fig. 2a, b). This suggests that NELF-mediated pausing occurs to some extent at virtually all genes transcribed by Pol II.

However, decreased Pol II density in NELF-depleted cells is not restricted to promoter-proximal regions, as Pol II levels are also generally decreased within gene bodies (Fig. 1b right, Supplementary Fig. 2a–c). While these decreases in Pol II density are widespread, significant, and reproduced across multiple replicates and experiments (see later section), the spike-ins indicate that this global phenotype is not generally recapitulated at the mRNA level (Fig. 1c). Compared to PRO-seq, we find far fewer significant changes in mRNA-seq following NELF depletion (Fig. 1d), although the genes that do change often do so in accordance with the changes observed in PRO-seq (Fig. 1e, Supplementary Fig. 2d). While differences in sensitivity between the two assays likely explain some of these discrepancies, the spike-in results from the two assays are derived from identical input material and are robust to many sources of variance in gene-

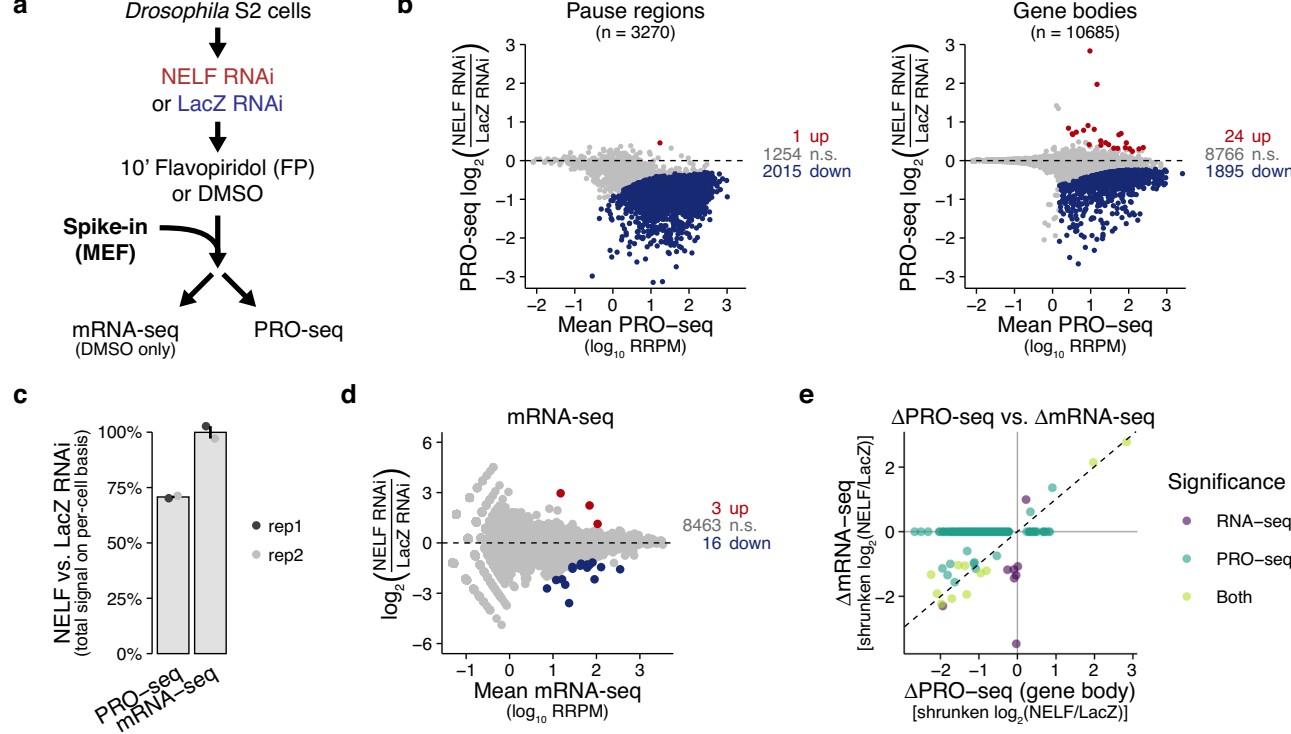

**Fig. 1 | NELF-depleted S2 cells have substantially reduced pausing but generally maintain gene expression. a** *Drosophila* S2 cells were treated with dsRNA targeting LacZ (control) or NELF (NELF-E) for a total of 4 days before a 10' treatment with either DMSO or the Cdk9 inhibitor Flavopiridol (FP). Cells were counted with spectrophotometry before the addition of exogenous spike-in material (mouse embryonic fibroblasts) and this material was divided for use in PRO-seq, RNA-seq (DMSO-only), and Western blotting assays (DMSO-only). FP results analyzed in subsequent figures. **b** MA plots showing changes in NELF vs. LacZ RNAi for PRO-seq reads within PRO-cap filtered pause regions (left) and consensus gene body regions (right), colored by DESeq2 significance (n.s. is non-significant). Mean read counts (x-axes) are based on Relative (spike-in adjusted) Reads Per Million mapped reads (RRPM), and y-axes use log fold-change "shrinkage" (from R package *apeglm*) to generate statistical estimates of log fold-changes such that low confidence

(high variance) measurements are adjusted toward zero (see Methods, Supplementary Fig. 2a, b). **c** Spike-in normalized read counts obtained for each assay represent the total material obtained on a per-cell basis. Spike-in normalized counts for NELF RNAi are being compared to their replicate-matched LacZ RNAi controls (mean ± SE) for n = 2 biological replicates. **d** MA plot for RNA-seq comparing NELF RNAi vs. LacZ RNAi, with points colored by DESeq2 significance. Spike-in normalization is used despite having no practical consequence in this case. **e** Comparison of quantitative changes in active transcription vs. stable mRNA after NELF RNAi. Shrunken log-fold changes in PRO-seq gene body regions (x-axis) are plotted against those measured in RNA-seq (y-axis) for a matched set of genes. Only genes with DESeq2 significance in at least one assay are plotted. Most genes are on the x-axis line. Dashed line represents y = x.

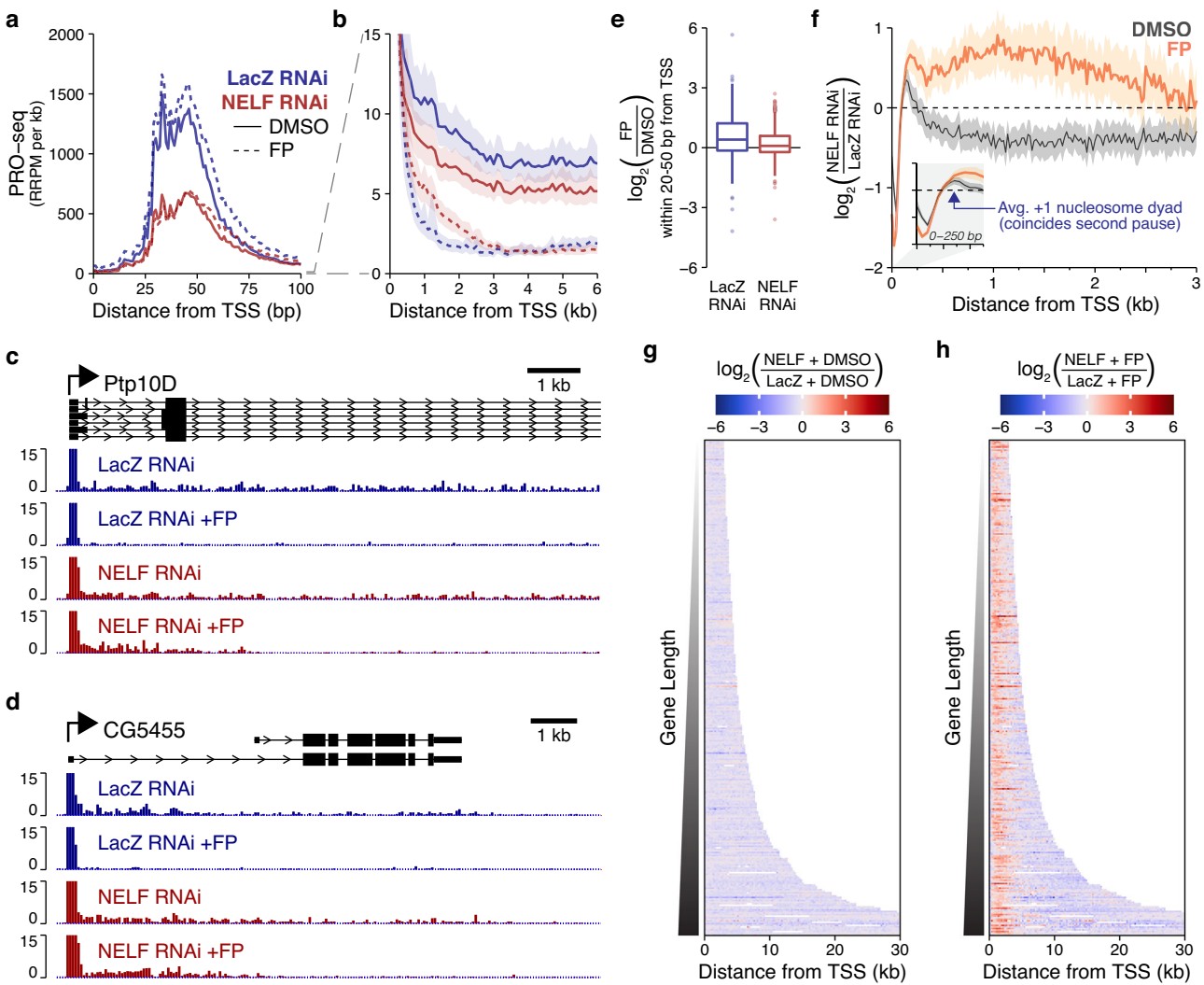

**Fig. 2 | NELF-depleted cells have aberrant early gene body Pol II density following Cdk9 inhibition. a** Metagene profiles of spike-in normalized PRO-seq signal within pause regions (mean signal at each single-base position) in units of Relative Reads Per Million mapped reads (RRPM, see methods), multiplied by 1000 (RRPM per kb). Pause regions are restricted to sites near annotated TSSes with focused sites of initiation lacking upstream transcription ($n = 3280$). Bootstrapping is used (as in b), but confidence intervals are removed for clarity (see Supplementary Fig. 3a, b). **b** Metagene profiles of PRO-seq (bootstrapped mean + 50% confidence interval) over gene body regions (TSS to TSS + 6 kb) in 100 bp bins. Binning is accounted for in calculating RRPM/kb. Genes are >6.8 kb and have focused sites of initiation lacking upstream readthrough transcription ($n = 469$). Plotting in the pause region is truncated by the y-axis limit. Dotted line connecting panels a-b only

provided to emphasize difference in y-axis. **c, d** PRO-seq (RRPM/kb) is shown for two example genes. Pause regions are not visible due to y-axis scaling. For Ptp10D, the first 10 kb are shown. **e** PRO-seq shrunken log2 fold-changes within the canonical pause regions of the same genes used in panels a-b ($n = 3280$ genes). Boxes show median and interquartile range (IQR), whiskers show 1.5x IQR. **f** For each drug treatment condition, log2 ratio of NELF to LacZ RNAi PRO-seq signal is calculated within 20 bp bins for each gene plotted in b. Bootstrapped means +75% confidence intervals are plotted. Arrow (inset) also marks the "second pause". **g–h** log2 fold-change in NELF vs. LacZ RNAi in DMSO **g** and FP **h** conditions are calculated within 250 bp bins. To maintain single-gene resolution in the heatmaps, only filtered genes longer than 3 kb with greater than 10 RRPM/kb between TSS + 500 and CPS – 500 in LacZ RNAi are shown ($n = 319$).

specific expression. Additionally, the lack of a global change in total mRNA abundance is also consistent with the lack of any obvious defect in cellular growth (see Methods). While a global change in Pol II elongation rates would explain a discrepancy between Pol II density and actual transcriptional output, we rule out this possibility in a later section. Rather, it is possible that the decrease in gene body transcription is not sufficient to substantially affect the pool of stable mRNA, potentially due to a global feedback stabilization of mRNA[30–35].

We conclude that the NELF-depleted S2 cells used in our experiments mostly maintain gene expression and cellular homeostasis despite substantial reductions in pausing across virtually all Pol II promoters.

## NELF-depleted cells have impaired responses to Cdk9 inhibition at sites of pausing and elongation

In both mammals and *Drosophila*, Cdk9 inhibition is known to prevent release of paused Pol II into the gene body, which causes an increase in Pol II occupancy at pause sites, while continued transcription by elongating Pol II results in a progressive clearing of the gene body[2,36–42]. Consistent with this, treatment of our LacZ RNAi controls with FP results in increased Pol II density within the pause region, while levels of elongating Pol II within the early gene body region are sharply reduced (Fig. 2a–d blue, Supplementary Fig. 3A).

By contrast, Cdk9 inhibition in NELF-depleted cells induces far less accumulation of Pol II density within pause regions (Fig. 2a, e, Supplementary Fig. 3a). Simultaneously, significant polymerase

density remains in the first 3-4 kb of the gene body, despite appearing to clear normally from positions further downstream (Fig. 2b–d, Supplementary Fig. 3d). This abnormal enrichment of early gene body Pol II is apparent across a large fraction of our gene list, and in almost all cases this density is largely restricted to the first several kilobases of the gene body (Fig. 2g, h, Supplementary Fig. 3e).

One possible explanation for this early gene body enrichment is Pol II "leaking" from the pause region into sites immediately downstream—a phenotype that has been previously associated with NELF-depletion[6,13,43]. Indeed, in the NELF-depleted DMSO condition, Pol II density within the pause region is generally shifted more downstream, and this phenotype is only exacerbated following Cdk9 inhibition (Fig. 2a, Supplementary Fig. 3c). However, unlike those more localized shifts within the pause region, the early gene body enrichment of Pol II (which extends several kilobases into the gene body in the NELF RNAi +FP condition) is not present in the NELF RNAi +DMSO condition, indicating that abnormal accumulation of Pol II within that region only occurs in the absence of Cdk9 activity (Fig. 2b, g, Supplementary Fig. 3d).

Immediately downstream of the canonical pause is a transitional area (-100–500 bp from the TSS) in which Pol II density declines sharply (often >100-fold in our S2 cells) (Fig. 2b). To visualize changes in this region, we calculated for each gene the changes in NELF vs. LacZ RNAi cells within bins, and we aggregated these into a metagene profile (Fig. 2f). Comparing the two DMSO conditions, the decrease in Pol II density at the canonical pause in NELF-depleted cells is accompanied by a local increase in Pol II density immediately downstream. This matches a previous finding that acute NELF depletion causes a downstream shift in promoter-proximal Pol II density into a defined "second pause" region in which the active site of Pol II appears roughly coincident with the +1 nucleosome dyad[6]. While more subtle in our experimental system, we find that our "second pause" also coincides with the average dyad position of the +1 nucleosome (Fig. 2f inset), which in *Drosophila* averages ~140 bp from the TSS[44].

Interestingly, Pol II continues to shift into (and accumulate within) the "second pause" following Cdk9 inhibition (Fig. 2f yellow). This indicates that NELF makes the release of Pol II from the canonical pause Cdk9-dependent, but that in the absence of NELF, Cdk9 still promotes Pol II overcoming the barrier at the second pause. However, the more localized "second pause" present in NELF RNAi (±FP) remains notably distinct from the much-expanded domain of abnormal Pol II density seen within the early gene body only after Cdk9 inhibition. This suggests that these densities represent distinct populations of Pol II complexes.

Together, these results indicate that NELF maintains Pol II within the canonical pause region (25–50 bp from the TSS) until Cdk9 mediates pause-release. In the absence of NELF, Pol II can move past the canonical pause without Cdk9, but this can result in stalling at another site further downstream (100–200 bp from the TSS). While our results in the FP condition indicate that Cdk9 activity also promotes Pol II overcoming this downstream barrier, the additional gene body phenotype (the abnormal density ~300–3000 bp from the TSS) suggests that the second pause does not functionally substitute for a NELF-mediated pause, which is required to prevent that accumulation in the absence of Cdk9 activity.

### More highly paused genes maintain more aberrant Pol II density in the early gene body in NELF RNAi +FP

In the previous section, we noted that in NELF-depleted cells, most genes abnormally maintain Pol II within the early gene body following Cdk9 inhibition. While the general pattern of this abnormal density is broadly consistent across genes transcribed by Pol II, the quantitative severity of this phenotype varies considerably (Fig. 2h, Supplementary Fig. 3e). Looking more closely, we found that the level of promoter-proximal pausing at a given gene is strongly associated with the quantity of that aberrant density.

To reach this conclusion, we first quantified (at individual genes) the change in PRO-seq density within the early gene body following FP treatment, and we plotted those changes against the starting (DMSO) density within: (a) that same region, or (b) the pause region immediately upstream (Fig. 3a). In both NELF and LacZ RNAi, the starting density within the early gene body was a generally poor predictor of the change following FP (Fig. 3a left). In contrast, pause region PRO-seq positively correlates with higher retention of PRO-seq signal within the early gene body following FP treatment in NELF-depleted cells (Fig. 3a right). This suggests that this abnormally persistent Pol II density has more to do with pausing (i.e., the amount of Pol II upstream before Cdk9 inhibition) than it does with the extent of gene transcription more generally.

While the plots in Fig. 3a effectively show the direct quantitative relationships between Pol II densities before vs. after 10′ FP treatment, the predictors (x-axes) do not address the important lurking relationship between pause density and gene body transcription, and the response variable (y-axis) does not isolate the most relevant phenotype, which is how the response to Cdk9 inhibition differs in NELF vs. LacZ RNAi cells.

To address these limitations, we defined phenotypic severity as the shrunken log2 fold-change (see Methods) in NELF RNAi +FP vs. LacZ RNAi +FP (as in Fig. 2f, h). Comparing the two FP conditions avoids conflating the FP-response phenotype (gene body clearing) with the already-decreased Pol II density observed in NELF-depleted cells (Figs. 1b and 2b) while also controlling for the expected level of gene body clearing (which may vary by gene). For each gene, we compared this measure of phenotypic severity against the average basal (DMSO) Pol II densities within both the pause and early gene body regions in something like a "3D MA plot" (Supplementary Fig. 4a). To address over-plotting, we divided the genes into bins to group the results (Fig. 3b, Supplementary Fig. 4b). This analysis confirms that this gene body enrichment phenotype is primarily and robustly associated with pausing level.

Finally, we generated PRO-seq metagene profiles to visualize Pol II density across genes divided by pause density (Fig. 3c). For this, we selected a set of genes with limited variation in gene body density, but large variations in pause density, and little correlation between the two (Supplementary Fig. 4c, d). While genes with higher pausing enrich more Pol II within the early gene body in NELF RNAi +FP, the overall pattern of this enrichment is similar for all pausing classes, consistent with the global nature of this phenotype.

We have additionally verified that a gene's response to NELF-depletion (DMSO condition) has little relationship to the quantity of abnormal Pol II density observed following FP treatment (Supplementary Fig. 4e, f).

Our analyses here show that in NELF-depleted cells, the abnormal Pol II density seen within the early gene body after FP treatment is not predicted by the density of Pol II that had been in that region before FP treatment, but instead by the Pol II that had been in the upstream pause region. This would be expected if those Pol II complexes were "leaking" downstream in the absence of NELF and Cdk9. Interestingly, highly paused genes, which have much higher initiation rates than pause release rates, still have sufficient Cdk9 to prevent the abnormal accumulation of Pol II in NELF RNAi (DMSO) cells; it's only when both NELF and Cdk9 activity are reduced that we see the abnormal early gene body density (Figs. 2f, g and 3c).

### Without NELF, Pol II can enter the gene body without Cdk9, but is unable complete gene transcription

We reasoned that many of our observations above could be explained if these NELF-depleted *Drosophila* cells were responding to Cdk9 inhibition in the same manner as the NELF-lacking fission yeast *Schizosaccharomyces pombe*, which we previously characterized[25]. In that study, we used PRO-seq to monitor transcription at various time points

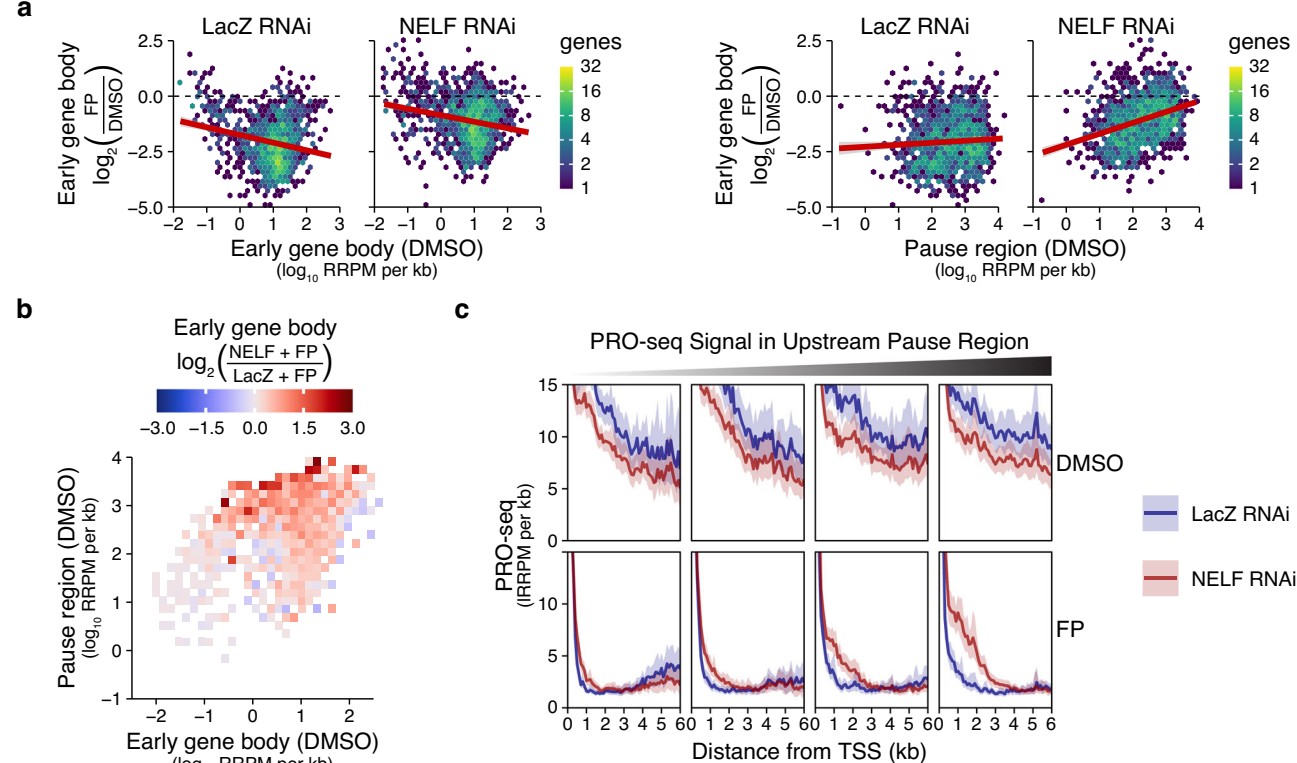

**Fig. 3 | Aberrant gene body Pol II accumulation is associated with more highly paused genes. a** For each RNAi condition, the PRO-seq (non-shrunken) log2 fold-changes within early gene bodies (300–1300 bp from the TSS) following FP are plotted (y-axes) against the DMSO PRO-seq signals either (left panels) within those same early gene body regions, or (right) within the pause regions (0–100 bp from TSS) immediately upstream. Filtered genes >1.6 kb are plotted ($n = 1243$). Linear regression lines (red) are shown with 95% confidence intervals. **b** For the same genes as panel a, 3-dimensional MA plot showing the relationship between pause region and gene body PRO-seq densities in DMSO conditions (x and y axes) and the

log2 fold-change in NELF RNAi +FP vs. LacZ RNAi +FP (tile color). To avoid over-plotting, genes are grouped into 2D bins (x and y axes) and tile colors show the mean shrunken log2 fold change for each group of genes. **c** Metagene profiles of gene body PRO-seq (bootstrapped mean + 50% confidence interval) divided into quartiles according to pause region (0–100 bp from TSS) PRO-seq signal. Genes from panels a-b were further subset to obtain a set of genes that represent a wide-range of pausing levels while having roughly similar Pol II densities within the early gene body region ($n = 492$, Supplementary Fig. 4c, d).

following Cdk9 kinase inhibition, and we found that in *S. pombe*, unlike in mammals or *Drosophila*, Pol II continues to enter the gene body in the absence of Cdk9 activity, but that these polymerase complexes have reduced elongation rates – a phenotype consistent with a critical role for Cdk9 in elongation complex assembly[20,45–47]. If our NELF-depleted *Drosophila* cells were recapitulating the *S. pombe* response, this would indicate that the presence of NELF in *Drosophila* prevents gene body entry in the absence of Cdk9 activity.

To test our hypothesis, we again utilized a time-course inhibition of Cdk9 kinase activity followed by PRO-seq (Fig. 4a). In contrast to the typical *Drosophila* response to Cdk9 inhibition (in which Pol II accumulates within pause-regions while gene body Pol II exits in a "clearing wave"), we hypothesized that an additional "advancing wave" would be observed in NELF-depleted cells as a result of continued gene body entry by Pol II, followed by slower than normal elongation (Fig. 4b).

For this second experiment, we again verified depletion of NELF (Supplementary Fig. 5a, Supplementary Fig. 7b). Compared to our first experiment, spike-in quantification of PRO-seq libraries show a similar, but slightly greater decrease in global levels of engaged Pol II (Supplementary Fig. 5b, c). PRO-seq replicates are again highly correlated (Supplementary Fig. 5d).

For very long genes, progressive gene body clearing can be visualized for all timepoints (Fig. 4c). The extent of clearing is broadly in line with previous studies estimating average gene body elongation rates ~2 kb/min[36,48,49], and we find no indication that NELF depletion has appreciably altered Pol II clearing on the largest scale (Fig. 4e, f,

Supplementary Fig. 5f). Thus, at least for Pol II complexes that continue transcribing into distal gene body positions, their elongation rates are unperturbed by NELF depletion alone.

Within the early gene body regions, we again see that NELF-depleted cells maintain elevated Pol II density following FP treatment (Fig. 4d). Unexpectedly, however, the bulk of this aberrant Pol II density remains solely within the first 3-4 kilobases of the gene body throughout the time-course, as opposed to progressively shifting into more downstream positions as we had predicted (Fig. 4g, h, Supplementary Fig. 5e). But even while most of these Pol II complexes appear to be restricted to that early gene body region, the quantity of Pol II within that region increases throughout the time-course (Fig. 4d, f, g, Supplementary Fig. 5e, f). This indicates that within the NELF-depleted cells, Pol II continues to enter the gene body in the absence of Cdk9 activity, but that those complexes have processivity defects and are unable to advance beyond the early gene body.

### NELF-depleted S2 cells closely recapitulate the *S. pombe* response to Cdk9 inhibition

We were initially doubtful that these NELF-depleted *Drosophila* cells were phenocopying the *S. pombe* response to Cdk9 inhibition[25], given the lack of an obvious advancing wave (Fig. 4). However, a direct comparison of the two datasets reveals that there are in fact striking phenotypic similarities between the two systems (Fig. 5).

The *S. pombe* experimental system has the advantage of a genetically engineered analog-sensitive variant of Cdk9, which can

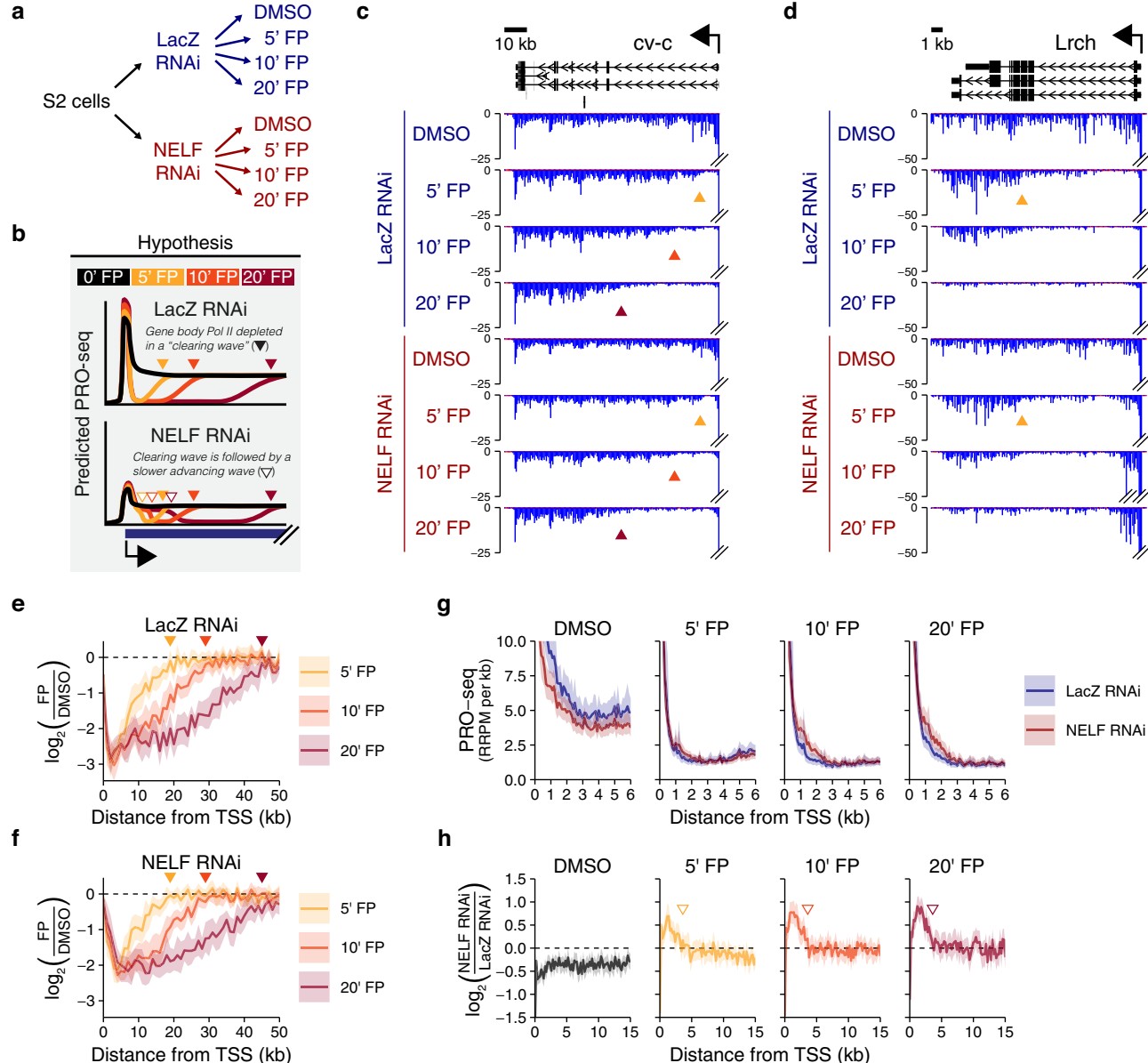

**Fig. 4 | Pol II entering the gene body in the absence of Cdk9 is largely restricted to the first 3-4 kilobases. a** A schematic of a Flavopiridol time-course that was performed on *Drosophila* S2 cells in the presence or absence of NELF knockdown (as in Fig. 1). Spike-in normalized PRO-seq was again used to accurately quantify the effects on active transcription. **b** Illustration of hypothetical PRO-seq profiles (not to scale). In both NELF knockdown and the LacZ control, we predicted that a "clearing wave" of gene body Pol II would be observed over the time-course, but that the NELF knockdown would also permit a slower "advancing wave" similar to what has been observed for the fission yeast *S. pombe*, which lack NELF. **c**, **d** PRO-seq (RRPM/kb calculated within 200 bp bins) at representative genes. The long gene cv-c **c** demonstrates the full extent of clearing waves, while the shorter gene

Lrch **d** is entirely cleared during the time-course. Double lines indicate signal exceeding the y-axis limit. Arrows provide matching points of reference. **e**, **f** For each RNAi condition, metagene profiles of PRO-seq log fold-changes (within 1 kb bins, bootstrapped mean + 50% confidence interval) following FP are plotted for filtered genes >55 kb ($n = 39$). **g** PRO-seq metagene profiles (bootstrapped mean + 50% CI within 100 bp bins) for filtered genes >6.8 kb in length ($n = 469$). **h** For each drug treatment, metagene profiles (bootstrapped mean + 50% CI) of PRO-seq log fold-changes in NELF vs. LacZ RNAi are calculated in 200 bp bins for each filtered gene >16 kb ($n = 227$). Arrows mark TSS + 3.6 kb, the average extent of the aberrant density after 20' FP.

be inhibited with precise kinetics using high concentrations of the inhibitor drug without incurring off-target effects[50–52]. With this system, a coherent advancing wave is visible within 30 s of inhibiting Cdk9, and this wavefront clearly progresses at subsequent timepoints (Fig. 5 top). However, within 5 min, the profile of the advancing wave has essentially reached its final form: a broad peak modally centered ~1–1.5 kb downstream of the TSS and predominantly restricted to the first 3 kb of the gene body. In the NELF-depleted *Drosophila* cells, the aberrant Pol II density observed following Cdk9 inhibition is not only predominantly restricted to a

similar range within the early gene body, but also forms a broad peak of a similar profile and a nearly coincident apex (Fig. 5 bottom).

The close recapitulation of the *S. pombe* Cdk9 inhibition phenotype in NELF-depleted *Drosophila* cells demonstrates the distinguishing role that NELF plays among the otherwise highly conserved processes of early transcription. In either model system, Cdk9 kinase activity is essential for successful gene transcription by Pol II, but only with NELF is there a strict early checkpoint for Cdk9 activity that blocks gene body entry in its absence. As NELF stably maintains Pol II close to

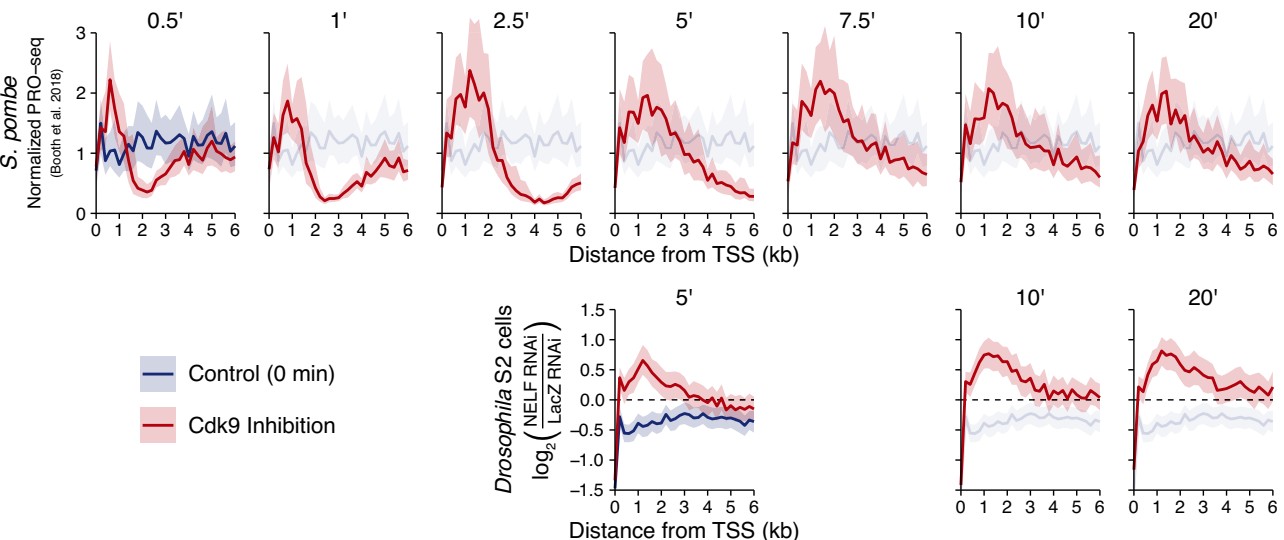

**Fig. 5 | The Cdk9 inhibition response of NELF-depleted *Drosophila* cells closely phenocopies that of the NELF-lacking fission yeast *Schizosaccharomyces pombe*.** PRO-seq profiles generated in *S. pombe* (top) at various times following Cdk9 inhibition (*n* = 42 length-filtered genes) from Booth et al. 2018, and the matching conditions from this study (bottom) highlighting the aberrant density in NELF vs. LacZ cells (as in Fig. 4h). Bootstrapped means + 50% confidence intervals are shown for 200 bp bins, and the untreated (no Cdk9 inhibition) condition is reproduced in each panel.

the TSS, this checkpoint also acts to prevent new rounds of initiation by Pol II[9,53].

While this manuscript was in preparation, a study was published demonstrating that rapid knockdown of the elongation factors Spt6 or PAF1 in human cells causes a transcriptional phenotype within the early gene body similar to what we observe following Cdk9 inhibition in *S. pombe* or NELF-depleted *Drosophila* cells[54]. We think these phenotypes are related, as a failure to load Spt6/PAF1 on Pol II is an expected downstream consequence of Cdk9 inhibition[20,46,55–57]. However, we find that the early gene body phenotype we both observe is essentially an elongation defect, which contradicts the conclusions reached in that study (see Discussion).

### Rapid turnover of engaged Pol II suggests defective elongation complexes undergo early termination

We next sought to determine why these Pol II complexes in NELF-depleted, Cdk9-inhibited cells are unable to transcribe beyond the early gene body. We hypothesized that these complexes are either a predominantly static population of "stalled" elongation complexes, or a dynamic, actively transcribing population that undergo multiple rounds of initiation and gene body entry followed by early termination.

To distinguish between these two possibilities, we built upon our previously established live-cell imaging system used to measure Pol II dynamics at a single locus[58]. This system uses dissected *Drosophila* (larval third instar) salivary glands, which contain giant polytene chromosomes[59,60]. These chromosomes are the product of 10 consecutive rounds of DNA replication without chromatid segregation, a process which results in ~1000 copies of each chromatid fiber remaining aligned lengthwise during interphase. The natural chromosomal amplification inherent to polytenes makes them uniquely advantageous for live cell imaging of single-locus protein kinetics, as individual genetic loci are present within large, visually distinct spatial environments[58,61–65].

The model locus is located within a gene desert and contains a single *Hsp70* transgene, as well as an upstream array of 256 LacO sites (Fig. 6a). The LacO sites are bound by ectopically expressed LacI-mCherry, which fluorescently labels the locus. In the absence of heat-stress, the *Hsp70* genes in *Drosophila* have very high levels of paused Pol II accompanied by modest levels of gene transcription[26,58,66,67]. These characteristics are common among the genes that most strongly

enrich abnormal gene body Pol II complexes in NELF RNAi +FP (Fig. 3), which makes the uninduced *Hsp70* gene a suitable model for this phenotype.

To measure Pol II turnover rates at the *Hsp70* model locus, we utilized a transgenic Pol II subunit (Rpb9) fused to a photoactivatable GFP (Rpb9-paGFP)[58]. paGFP is non-fluorescent until activated with a pulse of 405 nm light, at which point the turnover and diffusion kinetics of the activated population can be measured[68]. Critically, we use a 2-photon laser to gradually excite paGFP within a 3-dimensionally defined region over a relatively long period of 2–3 s, which avoids the activation of Pol II complexes that are freely diffusing or transiently interacting. Such complexes are also selected against during image acquisition (see Methods). Our imaging experiments thus measure the turnover kinetics of Pol II complexes that are stably engaged at the transgene (Fig. 6a).

We performed photoactivation experiments in salivary glands with RNAi knockdown of either NELF-D or Luciferase as a control, each with and without a >15 min treatment with FP (Fig. 6b, Supplementary Fig. 6a, Supplementary Fig. 7c, Supplementary Movies 1-4). Like NELF-E RNAi, NELF-D knockdown is also known to destabilize multiple NELF subunits[69]. As expected from our PRO-seq data, NELF-depleted glands have fewer Pol II complexes engaged at the Hsp70 transgene locus, as shown by the initial fluorescence intensity of photoactivated Pol II (Fig. 6c). To complement our imaging of Pol II itself, we also performed separate experiments to image nascent RNA production at the *Hsp70* transgene. This was done by using a fly line that expresses MCP-GFP (in place of Rpb9-paGFP), which binds to an array of 24 MS2 stem-loops formed in the intron of the nascent RNA of the *Hsp70* transgene (Fig. 6a). The significant reduction in local MCP-GFP fluorescence in the NELF-depleted glands confirms that reduced Pol II density is accompanied by reduced nascent RNA abundance (Fig. 6d, Supplementary Fig. 6b). While FP treatment in the Luciferase RNAi control significantly reduced both local Pol II and nascent RNA levels, its relative effect on the NELF-depleted glands was more subtle (Fig. 6c, d). The reduced transcription and weaker response to FP are broadly consistent with the effects of NELF depletion that we observed globally in S2 cells using PRO-seq.

Finally, we utilized the full time-course imaging of Rpb9-paGFP to determine Pol II turnover kinetics at the Hsp70 transgene in each condition (Fig. 6b, e, Supplementary Fig. 6c). Relative to the total size

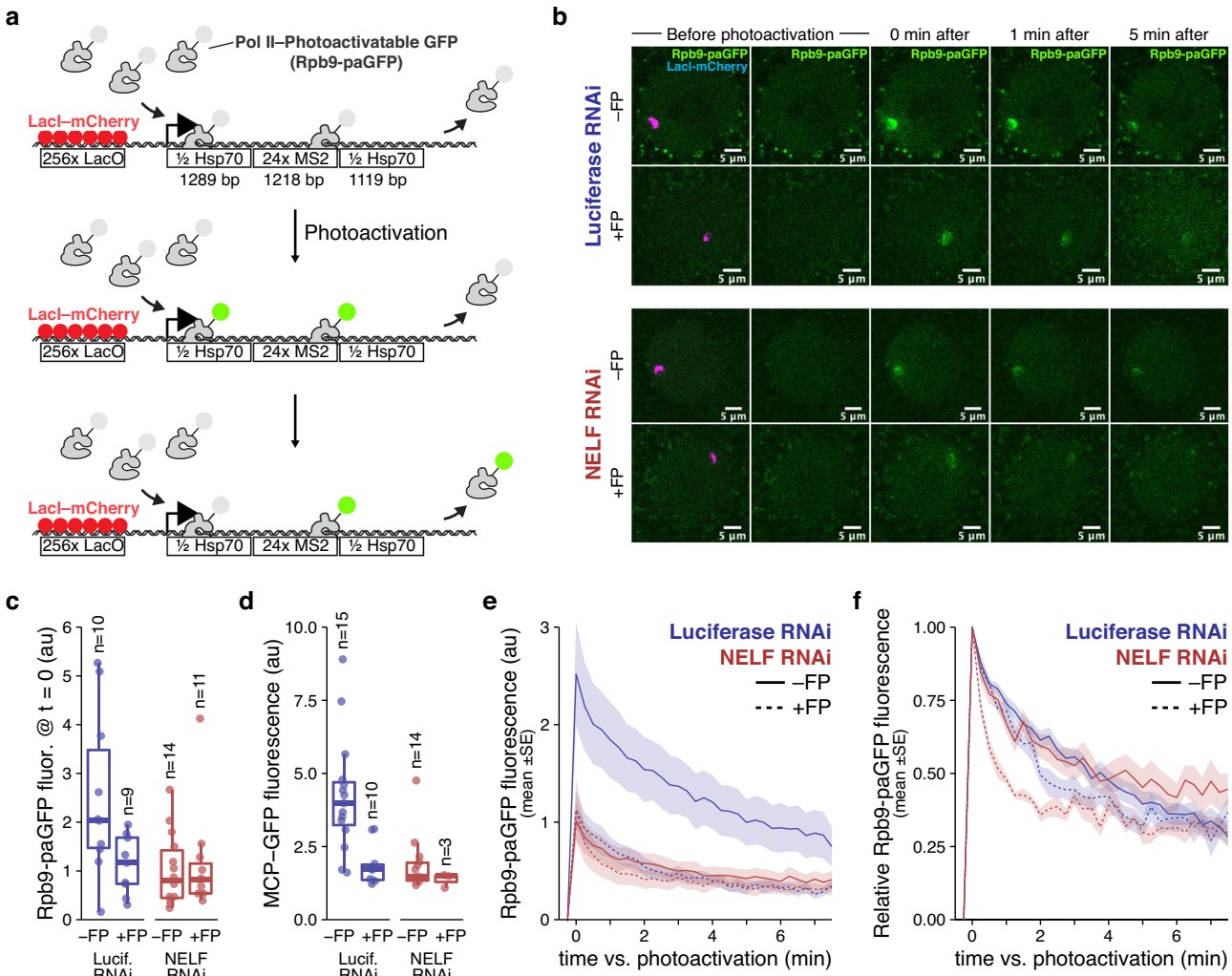

**Fig. 6 | In NELF-depleted *Drosophila* cells, Pol II engaged in the absence of Cdk9 activity is rapidly turning over. a** Schematic of live-imaging in *Drosophila* salivary glands using photoactivatable-GFP fused to Pol II (Rpb9-paGFP) at a single *Hsp70* transgene locus. Locus-specific photoactivation of Rpb9-paGFP is performed gradually with a 2-photon laser to avoid activation of unengaged/freely diffusing Pol II, and the turnover kinetics of the activated population are measured. The transgene locus is found using LacI-mCherry, which binds the upstream LacO sites. The MS2 sites within the transgene are utilized in separate experiments (**d**, Supplementary Fig. 6b). **b** Representative images showing Rpb9-paGFP before and after photoactivation for each RNAi condition ±FP (a > 15 min treatment; see Methods). For each condition, the locus with fluorescence intensities and relative turnover rates closest to the mean values are shown. **c** Rpb9-paGFP fluorescence intensity immediately after photoactivation (normalized to local pre-

photoactivation background fluorescence), which correlates with the size of the activated Pol II pool at the *Hsp70* transgene locus. Arbitrary units of fluorescence are consistent across conditions. Boxes show the median+IQR, whiskers extend to 1.5x IQR or to most extreme point if closer to median than 1.5× IQR. **d** Background-normalized MCP-GFP fluorescence intensity at the *Hsp70* transgene locus. MCP-GFP binds to an array of 24 MS2 stem-loops in the nascent transcript. Fluorescence intensity is a measure of the steady-state abundance of the nascent transcript. Boxplot statistics are the same as in **c**. **e** Rpb9-PAGFP fluorescence at the *Hsp70* transgene is plotted before and after local photoactivation (t = 0) for each condition. Data plotted in c corresponds to t = 0 (and units are identical). **f** For each individual photoactivation time-series, fluorescence intensity is normalized to be 0 just before photoactivation, and 1 immediately after (at t = 0). Source data are provided as a Source Data file.

of the photoactivated population, we found that Rpb9-paGFP in the NELF RNAi +FP condition decays noticeably faster than in any other condition (Fig. 6f, Supplementary Fig. 6d). In NELF RNAi +FP, >50% of locally engaged Pol II has turned over within the first minute—a fraction more than double that of any other condition. As these experiments capture single loci at a given point in time, we were unsurprised to find that the size of the photoactivated population varies across individual time-series (Fig. 6c). However, we verified that the high turnover rate in NELF RNAi +FP is robust to this variance (Supplementary Fig. 6d, e).

The accelerated turnover of chromatin-interacting Pol II in the NELF RNAi +FP condition is inconsistent with immobile Pol II "piling up" in the early gene body. And given that both NELF and Cdk9 are known to act after the initiation of transcription, the dynamic nature of these Pol II complexes is more in line with continued rounds of new

initiation followed by early termination. While these results have the caveat of correlating a phenotype at a single transgene in a dissected tissue to our PRO-seq results in cultured S2 cells, the global and reproduced nature of the early gene body phenotype in PRO-seq, and the various ways in which our live-imaging results match our expectations from PRO-seq, all support the view that these assays are characterizing the same phenotype.

## Discussion

In our summary model, we conclude that NELF prevents unproductive attempts at gene transcription from occurring in the absence of Cdk9 kinase activity by blocking Pol II from entering into the gene body (Fig. 7 bottom). In doing so, NELF enforces a strict, early checkpoint for Cdk9, whose kinase activity is essential for successful gene

Without NELF:

With NELF:

**Fig. 7 | A model illustration of a NELF-mediated checkpoint for Cdk9 activity.** The basic and essential functions of Cdk9 kinase activity during early gene transcription are highly conserved across eukaryotes. Without NELF, Pol II will continue to initiate gene transcription in the absence of Cdk9 activity, but the subsequent failure to form a competent elongation complex results in early termination. NELF prevents these unproductive rounds of transcription by blocking gene body entry when Cdk9 activity is insufficient, which creates an efficient means of controlling gene transcription through regulation of Cdk9. This strategy may be prominent but non-universal in higher eukaryotes, and NELF may be expendable in contexts where Cdk9 availability is not restricted.

transcription by Pol II in both fission yeast and *Drosophila*. This NELF-mediated checkpoint is a major point of functional divergence that stands in contrast to the functions of Cdk9 and the other general processes of early gene transcription by Pol II, which are otherwise highly conserved across distant eukaryotes. Without this NELF-mediated checkpoint, gene transcription by Pol II can still occur without measurable elongation defects (e.g., elongation rates are normal and Pol II is not enriched in early gene body regions) as long as Cdk9 kinase activity remains available (Fig. 7 top).

Interestingly, NELF depletion alone does not produce an accumulation of early gene body Pol II, as this phenotype also requires inhibition of Cdk9. This implies that the level of Cdk9 is sufficient to allow Pol II complexes that leak into the early gene body to achieve normal transcription elongation. This could be a consequence of known compensatory mechanisms that increase the general pool of active Cdk9 complexes in response to transcriptional perturbations[70–72]. However, while normal elongation remains possible in NELF-depleted cells, the widespread changes in transcriptional gene expression indicate that NELF plays a significant role in defining the transcriptional program of S2 cells (Fig. 1).

Given that NELF is essential for the efficient shutdown of gene transcription in response to a loss of Cdk9 activity, we propose that NELF would have been essential for the evolution of gene regulatory mechanisms centered around Cdk9 (or that NELF at least made such mechanisms tractable for widespread use). With NELF, Cdk9 availability can be strictly controlled—for example by inhibitory HEXIM/7SK complexes[45,73,74]—without the substantial inefficiency of globally failing gene transcription. If Cdk9 is the limiting factor (as we predict would be the case in highly paused, NELF-containing contexts), transcriptional gene expression could be substantially determined by the ability of regulatory factors to recruit and maintain Cdk9 activity near specific promoters. The provision of this regulatory strategy is one way that promoter-proximal pausing has been proposed to be functionally useful[10,75]. Our findings show how NELF enhances the regulatory

potential of Cdk9, but future work is necessary to better understand the conditions and extent to which gene transcriptional programs are determined through the regulation of Cdk9, and how that regulation is accomplished.

Interestingly, this also implies that NELF may not be essential when Cdk9 is readily available. This could explain how NELF has been lost multiple times in eukaryotic evolution[23,24], and may prove insightful for ongoing work characterizing the context-specific requirements for NELF during mammalian development[76]. However, other models describing the functional implications of pausing have been proposed that may contribute to a requirement for NELF even if Cdk9 is relatively available[75]. For instance, if NELF increases pausing half-lives even in the presence of high Cdk9 activity, that increased residence time might be useful for enforcing the position of the +1 nucleosome[14,77] or providing more time for capping to occur[6,78–80].

One critical and outstanding question is when and to what extent pausing is NELF-mediated vs. NELF-independent. Our results match a previous finding that the canonical pause (20–50 bp from the TSS) is NELF-mediated, and that without NELF, a new pausing event occurs downstream at the so-called "second pause"[6]. This "second pause" is reminiscent of the "proto-pausing" that we previously described in the fission yeast *S. pombe*[7]. However, the second pause does not create a strict requirement for Cdk9-mediated pause-release, and it does not occur when NELF is present. All of this supports the prevailing view that NELF-mediated pausing is a distinct form of pausing, and that its release is mediated by Cdk9.

A straightforward model is that NELF requires Cdk9 to activate Pol II before it exits the promoter-proximal region, and this ensures that elongation factors (EFs) are loaded before Pol II encounters nucleosomes in the gene body, which would otherwise cause a stall (the second pause). Thus, the absence of the second pause when NELF is present implies that EF loading occurs rapidly following (or coincident with) pause-release. Particularly insightful here are the recently published knockdowns of the EFs Spt6 and PAF1[54]. As stated earlier, these knockdowns appear to phenocopy Cdk9 inhibition in the absence of NELF, and are likely the relevant downstream consequence. The fact that these knockdowns produce this same phenotype when NELF is present implies that the NELF checkpoint enforces only a requirement for Cdk9, but not the downstream loading of these EFs (i.e., the EFs are not critical to release paused Pol II). Our interpretation disagrees with the authors' conclusion that depleting these EFs increases gene body density by increasing the rate of pause-release. The authors may have overlooked that Pol II density within the gene body will increase if elongation rates decrease, which is a known consequence of Spt6 perturbation[81–83]. An increase in cryptic initiation following Spt6 depletion may also contribute to increased gene body density[84–87]. Tellingly, in all examples shown in that recent paper[54], paused Pol II density appears generally unperturbed following EF depletion, which is inconsistent with increased rates of pause-release.

While our NELF-depletion recapitulated the Cdk9 inhibition response of *S. pombe*, a previous co-ablation of NELF and Cdk9 did not produce the same phenotype[6]. However, that experiment is distinct in that they first inhibited Cdk9 (shutting down gene transcription and stabilizing paused Pol II) for one hour before inducing rapid degradation of NELF. While our longer-term knockdown sacrifices sensitivity to the primary effects of NELF depletion (vs. downstream secondary effects), our approach has the benefit of measuring the immediate effects of Cdk9 inhibition in cells that are NELF deficient but that otherwise maintain cellular growth and functional gene expression. Altogether, our findings are a coherent extension of the bulk of the existing literature on pausing and Cdk9, which clarifies the distinct role played by NELF and points toward increased regulatory roles for Cdk9 in higher eukaryotes.

## Methods

### Preparation of dsRNA for RNAi knockdowns

For RNAi in S2 cells, dsRNA targeting either LacZ or NELF-E was generated through in vitro transcription (IVT) using T7 RNA polymerase (NEB M0251) and DNA templates flanked by T7 promoters. DNA templates were generated by PCR (primers in Supplement) from gDNA using a homemade Pfu-Sso7d DNA polymerase with Phusion® HF buffer (NEB) and purified using either PAGE or SPRI beads (Agencourt AMPure XP®) at a volumetric ratio of 1.6:1. IVT was performed overnight at 37 °C using 600–800 ng template in the manufacturer's reaction buffer supplemented with Superase-In™ (Invitrogen AM2694), 5 mM DTT, and yeast inorganic pyrophosphatase (NEB M2403). Reactions were treated with 2U DNase I (Ambion AM2222) for 30 min @ 37 °C before acid phenol-chloroform extraction and isopropanol precipitation. Resuspended dsRNA was melted for 3 min at 90 °C before re-annealing on ice for 3 min, and the final products were verified by gel electrophoresis.

### S2 cell culture, RNAi, FP treatment, and exogenous spike-in

*Drosophila melanogaster* S2 cells were grown at 26 °C in M3 + BPYE medium with 10% FBS. For RNAi knockdowns, 15 mL of cells at $1 \times 10^6$ cells/mL were seeded in FBS-free media and treated with dsRNA at 10 μg/mL for 45 min before another 15 mL of media with 20% FBS was added. After 2 days, cells were split into two flasks and treated a second time with dsRNA. After another 2 days, the two flasks were recombined and cells were counted using OD600 using conversion factor $0.1394 = 1 \times 10^6$ cells/mL (counts in Supplement). Cells were divided into tubes (~$1.9 \times 10^7$ cells each) for treatment (at 26 °C) with 500 nM Flavopiridol or solvent only (0.02% DMSO). Treatments were abruptly stopped by inserting the tubes into ice. S2 cells were pelleted 5 min at $1000 \times g$ (4 °C), resuspended in PBS, and $8 \times 10^4$ freshly thawed mouse embryonic fibroblasts (MEFs) were added as exogenous spike-in.

### Western blots for NELF-E knockdown in S2 cells

During cell counting, a 2 mL sample of each S2 cell culture was set aside in 1% (v/v) PMSF, pelleted at $1000 \times g$, and resuspended at a final concentration of $1 \times 10^5$ cells/μL in 1× SDS Loading Buffer (50 mM Tris-Cl pH 8.3, 2% (w/v) SDS, 167 mM DTT, 10% (v/v) glycerol, 0.1% (w/v) bromophenol blue) and boiled 5 min at 95 °C before SDS-PAGE (10%/tris-glycine) and wet transfer onto nitrocellulose membranes. Membranes were blocked 30 min in 2% BSA in TBST (TBS + 0.1% Tween® −20) and probed in that buffer using rabbit anti-NELF-E (1:1000; a gift from David Gilmour, Pennsylvania State University)[69,88,89] and guinea pig anti-Chromator (1:2000) which was previously made for and used by our lab[90]. Fluorescent secondary antibodies from LI-COR Biosciences (anti-rabbit IRDye® 800CW #926-32213 and anti-guinea pig IRDye® 680LT #926-68030) were used at 1:15000 in TBST before imaging on a LI-COR Odyssey®.

### S2 cell permeabilization for PRO-seq

Keeping S2 cells (with MEF spike-ins) ice-cold throughout, cells were pelleted and resuspended in 5 mL Wash Buffer (10 mM Tris-Cl, pH 7.5; 10 mM KCl; 150 mM sucrose; 5 mM MgCl$_2$; 0.5 mM CaCl$_2$; 0.5 mM DTT; 1× Protease inhibitor cocktail (Roche cOmplete™); 20U Superase-In™ (Invitrogen) RNase inhibitor) before resuspension in Permeabilization Buffer (10 mM Tris-Cl, pH 7.5; 10 mM KCl; 250 mM sucrose; 5 mM MgCl$_2$, 1 mM EGTA; 0.05% Tween®−20; 0.1% Nonidet® P-40 substitute (VWR M158); 0.5 mM DTT; 1× Protease inhibitor cocktail (Roche); 20U Superase-In™) and left on ice for 5 min. Cell permeability was assessed >99% under a microscope in the presence of 0.2% Trypan Blue. Permeabilized cells were washed twice in 5 mL Wash Buffer, resuspended in 220 μL Storage Buffer (50 mM Tris-Cl pH 8.3; 40% glycerol; 5 mM MgCl$_2$; 0.1 mM EDTA; 0.5 mM DTT), divided between 2 aliquots (~$9 \times 10^6$ cells each), flash frozen in liquid nitrogen, and stored at −80 °C.

### PRO-seq and PRO-cap library preparation

PRO-seq and PRO-cap libraries were prepared as described previously[27], with minor modifications. The 3′ adapter for the initial PRO-seq experiment contains a 6 N UMI (5′-/5Phos/NNNNNNGAUC-GUCGGACUGUAGAACUCUGAAC/Inverted dT/−3′) to remove PCR duplicates. The 3′ adapter for the time-course experiment contains library-specific barcodes in place of the UMI (5′-/5Phos/GNNNNNNGAUCGUCGGACUGUAGAACUCUGAAC-/Inverted dT/−3′). For S2 cell PRO-cap, tobacco acid pyrophosphatase (TAP) (EpiCentre T19500, discontinued) was used for the enzymatic removal of the 5′ cap structure. Just before this step, the sample was divided into two tubes to include a buffer-only (TAP-) control to identify background (uncapped) signal[91]. Libraries were PCR amplified with 11 cycles and PAGE purified before sequencing (1 × 75 bp) on an Illumina NextSeq® 500/550.

### mRNA-seq library preparation

RNA-seq libraries were prepared from the same material (S2 + MEF spike-in) used for PRO-seq. RNA was extracted from ~$1.9 \times 10^6$ cells using TRIzol™ LS/ethanol precipitation, treated 2U DNase I (Ambion AM2222) 1 h at 37 °C, followed by a second TRIzol extraction/ethanol precipitation. RNA was quantified using Qubit™ RNA BR assay, and 400 ng was used with the QuantSeq™ 3′ mRNA-Seq Library Prep Kit (FWD) (Lexogen A01172), which uses oligo-dT to prime for first strand synthesis and random priming for second strand synthesis. The manufacturer's UMI Second Strand Synthesis Module was used to add UMIs. Libraries were PCR amplified for 12 cycles, quantified (Qubit™ dsDNA HS assay), pooled, and sequenced (1 × 75 bp) on an Illumina NextSeq® 550.

### PRO-seq/PRO-cap processing and alignment

For PRO-seq and PRO-cap libraries, adapters were trimmed using cutadapt (v1.18) (−nextseq-trim 10−minimum-length 15). For PRO-seq libraries with UMIs, we used PRINSEQ[92] (prinseq_lite v0.20.4) to remove PCR duplicates before adapter trimming. Using bowtie2 (v.2.3.5) (−very-sensitive), we first aligned to rDNA repeats to deplete nascent rRNA before then aligning to a combined genome containing all chromosomes for *Drosophila melanogaster* (dm6) and *Mus musculus* (mm10). Because some S2 cell material can incorrectly map uniquely to mm10, we N-masked all sites in mm10 that S2 cell PRO-seq without spike-in[93] can incorrectly map. R/Bioconductor package BRGenomics (v0.8.0-v1.4.0) was used to remove and count reads mapping to the spike-in genome, and to select the proper sites for PRO-seq (the 3′ end of the nascent RNA before the biotin run-on) and PRO-cap (the 5′ end of the nascent RNA) and export bigWig files of single-base coverage.

### RNA-seq processing and alignment

For RNA-seq, fastp (v0.20.0) was used to trim adapters and extract UMIs for efficient, post-alignment deduplication using UMI-tools[94] (v1.0.0). Alignment to the same combined genome was performed using STAR aligner (v.2.7.3a) (−outFilterMultimapNmax 20−alignSJoverhangMin 8−alignSJDBoverhangMin 1−alignIntronMin 20−alignIntronMax 1000000). BRGenomics (v1.4.0) was used to extract and count spike-in reads and export bedGraph files containing full read-spans.

### PRO-seq and RNA-seq normalization

Spike-in normalization was performed with the R/Bioconductor package BRGenomics (v1.4.0) using its batch-normalized "SRPMC" (spike-in normalized reads per million in the control): within each replicate, LacZ RNAi +DMSO is depth-normalized as RPM, while the other conditions are in matching units "relative RPM" (RRPM) based on the spike-in. The final normalization factor for a sample, *i*, in terms of the reads mapped to dm6 and mm10 for that sample and for the

replicate-matched LacZ RNAi +DMSO:

$$NF_i = \frac{mm10_{LacZ.DMSO}}{mm10_i} \cdot \frac{10^6}{dm6_{LacZ.DMSO}}$$

In the first PRO-seq experiment (Figs. 1–3, Supplementary Figs. 1–4), there was higher replicate disagreement in the spike-in normalized Flavopiridol-treated conditions (Supplementary Fig. 1e), which was not an issue for the DMSO-treated conditions or in any condition in the second (time-course) experiment (Supplementary Fig. 5b). Calculating normalization factors for the FP conditions using reads at the ends of long genes (which, compared to the matching DMSO-treated sample, are not expected to have changed yet in response to FP) eliminated this replicate discrepancy in the first experiment (Supplementary Fig. 1f), while having no meaningful effect on any condition in the second experiment (Supplementary Fig. 5c). To be consistent, we used this normalization approach for the FP conditions in both experiments. While our findings (Supplementary Fig. 5b, c, Supplementary Fig. 1f) indicate that this adjustment improves accuracy, we have analyzed our data using only the spike-in normalization and it does not substantially alter our findings.

### Adjustment of PRO-cap using the TAP- control
We used the TAP- PRO-cap condition to identify sites from which uncapped molecules may non-specifically enter a PRO-cap library. However, we found that the TAP- condition is far more diffuse than the TAP+ condition and that TAP- is a poor predictor of TAP+ signal. To penalize some of these TAP- sites without overcorrecting, we used the proportion of reads mapping to rDNA as an estimate of the extent to which TAP- was enriching non-capped molecules vs. the TAP+ condition (-3.14x). We then down-scaled the TAP- by this factor before subtracting this signal from the TAP+ PRO-cap data.

### Gene annotation adjustment and filtering
For all details, see Supplemental Methods. In brief, PRO-cap was used to adjust TSS annotations and to select only those that locally concentrate signal and which lack upstream readthrough transcription in PRO-seq. For analyses requiring matched promoters and gene bodies, we strictly filtered for genes where these are unambiguously matched ($n = 1560$). For some differential expression analyses, we used all filtered TSSes ($n = 3280$) for promoter-proximal regions, and all gene bodies >100 bp after taking the most downstream annotated TSS + 300 bp and upstream CPS − 300 bp ($n = 12929$). For RNA-seq, we used only regions within 500 bp upstream of an annotated CPS for genes included in the $n = 12929$ gene body regions above ($n = 12876$).

### Browser shots and metagene profile plots
Browser shots were generated using R/Bioconductor package BRGenomics (v1.4.0) via this Github script (https://github.com/mdeber/browser_shot_plotter). Metagene profiles were generated using the default parameters of BRGenomics function metaSubsample: 1000 iterations of randomly sampling 10% of the genes and calculating the mean signal at each position or bin. The bootstrapped mean is the 50th percentile of the sample means, and other quantiles are used for the confidence interval. Normalized signal within a bin is divided by the bin size and multiplied by 1000 to convert to units RRPM/kb. For metagene profiles of log2 fold-changes (LFCs), the LFCs are calculated directly within bins for each individual gene, and those LFCs are subjected to the procedure above.

### Differential expression analyses
DESeq2 (v1.32.0) was utilized indirectly via BRGenomics (v.1.4.0) to use comparison-specific dispersion estimates instead of DESeq2's default sample-blind estimates, which makes the comparisons truly pairwise as samples not being compared do not contribute to the dispersion

estimates. This is important because of the various global changes induced by our perturbations. To incorporate spike-in normalization, we specified DESeq2 sizeFactors by inverting our normalization factors before dividing those by their collective mean, which is necessary because DESeq2 sizeFactors are centered near 1 and their magnitude affects the significance test.

### Log fold-change shrinkage
Where noted, we use shrunken log2 fold-changes (LFCs) generated by apeglm[95] (v1.14.0) via DESeq2 via BRGenomics, following the procedures described in the previous subsection. LFC shrinkage conflates "low confidence/high variance" with "no change", which is a statistically useful estimate for an LFC whenever an isolated parameter estimate is required. We find these work very well with well-sequenced, highly reproducible PRO-seq libraries, having minimal effect on high-confidence LFCs while quantitatively de-emphasizing lower-confidence changes.

### Two-dimensional gene binning and "3D MA" plots
To divide genes into 2-dimensional bins of promoter PRO-seq and gene body PRO-seq, for each metric we made 30 evenly spaced bins in log10 space (Supplementary Fig. 4b). For the "3D MA" plot (Fig. 3b), the average shrunken LFC was calculated for all genes within a given bin using the BRGenomics function aggregateByNdimBins.

### Re-analysis of *S. pombe* PRO-seq data
PRO-seq data from *S. pombe* with a genetically engineered, analog-sensitive Cdk9 (Cdk9AS) was downloaded as bigWig files from GEO GSE102308, and the replicate-combined, spike-in normalized files were used as in the original publication[25]. The same filtered gene list was used as well.

### Fly lines obtained for this work
From the Bloomington Drosophila Stock Center, flies expressing: a larval salivary gland-specific Gal4 (c728-Gal4; stock #6983); a fusion MCP-GFP under a constitutive hsp83 promoter (hsp83-MCP-GFP; stock #7279); and UAS-RNAi$_{Luciferase}$ (stock #31603). UAS-RNAi$_{NELF-D}$ flies (*NELF-Di*[8-2] in the original publication[14,69]) were provided by David Gilmour (Penn State). We have previously described[58] the generation of the following 3 fly lines: UAS-Rpb9-paGFP; the line expressing an mCherry-LacI fusion from the minimal Sgs3 promoter; and the line containing the *Hsp70* transgene (256xLacO-Hsp70MS2; Supplementary Data 1).

### Fly lines generated for this work
Flies containing 256xLacO-Hsp70MS2 were crossed to those containing mCherry-LacI and recombinant progeny containing both transgenes (on chr2) were selected. Recombinant flies were also selected after crossing c728-Gal4 flies to UAS-Rpb9-paGFP flies (on chr3). Those recombinant lines were crossed to one another to generate:

$$Line\ 1: \frac{256xLacO-Hsp70MS2, mCherry-LacI}{CyO}; \frac{c728-Gal4, UAS-Rpb9-paGFP}{Tm6b, Tb}$$

In lieu of UAS-Rpb9-paGFP, another recombinant line was generated that contains c728-Gal4 and MCP-GFP on chr3, which was crossed to the Hsp70 + mCherry-LacI recombinant flies to generate:

$$Line\ 2: \frac{256xLacO-Hsp70MS2, mCherry-LacI}{CyO}; \frac{c728-Gal4, MCP-GFP}{Tm6b, Tb}$$

### Salivary gland dissection
Two male flies from line 1 or line 2 generated above were crossed to 5–8 females with UAS-RNAi$_{NELF-D}$ or UAS-RNAi$_{Luciferase}$. Salivary glands

were dissected from non-tubby wandering third instar larvae and 3–5 were placed into a MatTek® Glass bottom dish (coverslip untreated No 1.5) containing 250 µL Grace's Insect Medium. Glands were gently placed in solution and the fat pads were gently tapped down to hold the glands in place. Care was taken not to touch the glands with forceps. A No 1.5 coverslip was placed over the entire well to limit motion. Flavopiridol treatments (500 nM) were done in Grace's medium, allowing 15 min for FP to enter the glands.

### Western verification of NELF knockdown in salivary glands

UAS-RNAi$_{NELF-D}$ or UAS-RNAi$_{Luciferase}$ flies were crossed with c729-Gal4 flies, and 20–30 pairs of salivary glands were dissected (as above). Glands in 50 µL Gland Dissection Buffer (100 mM Tris-Cl, pH 7.5; 150 mM NaCl; 10 mM EDTA; 1% SDS; 1× Protease Inhibitor Cocktail (Pierce A32963)) were sonicated in a bath sonicator (Diagenode Bioruptor® UCD-200 TM) on High for 5 min, 30 s on/30 s off. Protein was quantified using Qubit™ Protein Assay (Invitrogen Q33212). 30 µg aliquots were put into the same Loading Dye and procedure as described for S2 cells, with the following changes. SDS-PAGE was done with a 4-20% gradient gel before transfer onto PVDF. After blocking for 30 min in 5% non-fat dry milk in PBST (PBS + 0.1% Tween®−20), membranes were cut below the 100 kDa marker before separately probing with primary antibodies guinea pig anti-dSpt5C (1:1000; previously raised for us against residues 732–1054 by Pocono Farms)[96] and rabbit anti-dNELF-D (1:1000; David Gilmour, Pennsylvania State University)[69,88,89] for 2 h at room temperature. Secondary probes from LI-COR Biosciences IRDye® 800CW (donkey anti-guinea pig #926-32411 and anti-rabbit #926-32213) were used at 1:10000 in PBST.

### Microscopy equipment

Imaging experiments were carried out on a Zeiss LSM880 Laser Scanning Confocal Microscope with 405, 488, and 561 nm lasers and a SpectraPhysics Insight® tunable multi-photon laser (680–1300 nm) using a Zeiss C-Apochromat 40 × 1.20 NA water Korr FCS M27 objective and a GaAsP detector. Emission from 488 nm excitation was captured within 410-553 nm; emission from 561 nm excitation was measured within 588–696 nm. Imaging procedures were performed using Zeiss Zen Black software.

### Imaging paGFP fluorescence decay after photo-activation (FDAP)

Salivary glands were located in brightfield before the 561 nm laser was used to locate the *Hsp70* transgene, which is visible in ~50% of progeny (those that received mCherry-LacI and not CyO). This search process took 3–15 min. An elliptical ROI was drawn tightly around the locus, and a 6-11 slice z-stack was taken using 488 nm (at 0.5% power, measured at 105 µW for a calculated power-per-area of 59.2 kW/cm$^2$) and 561 nm (0.5% power, 20 µW or 8.5 kW/cm$^2$). Each slice takes ~1 s to collect. Photo-activation was performed with the SpectraPhysics Insight® multi-photon laser at 810 nm (2× the standard 405 nM). Activation was directed to the center of the z-stack and performed by scanning the region of interest 50× at 5% power (19 mW = 7712 kW/cm$^2$). Photo-activation took approximately 2–3 s depending on the size of the ROI. This longer activation time allows unbound molecules to diffuse out of the region of interest before the first z-stack is taken immediately after. Subsequent stacks are taken every 15 s.

### FDAP data processing

Z-stacks were converted into maximum intensity projections using ImageJ2. To quantify local Rpb9-paGFP signal, an ROI was drawn around the locus (marked by mCherry-LacI) and the intensity of Rpb9-paGFP fluorescence was measured using the ImageJ2 ROI manager. For background subtraction, the same sized ROI was moved to a random location in the nucleus and the 488 channel intensity was measured and subtracted from the paGFP signal at the *Hsp70* locus.

### FDAP bleach correction

A separate imaging experiment was performed in which whole-cells were photo-activated using 405 nm single-photon (1 P) laser excitation (which affects the entire depth of the cell) followed by acquisition of 6–20 slice z-stacks. A maximum intensity projection was made and the fluorescence of the entire nucleus was measured as a function of the number of laser scans. The natural log of fluorescence intensity was plotted against the number of scans and the resulting linear fit gave a constant $0.100 \pm 0.029\%$ signal bleached per scan.

### FDAP data plotting

When using paGFP signal as a relative proportion to the post-photoactivation signal, fluorescence intensity (background-subtracted and bleach-corrected) measured at the locus immediately prior to photoactivation was treated as the zero value, and the intensity immediately after photoactivation was treated as 1. Line plots of the average values (bleach corrected signal or relative intensity) were generated by taking the mean value for all samples in that condition, and the lower and upper bounds of the shaded regions denote 1 standard deviation from the mean.

### Locus-specific MCP-GFP imaging

Flies containing the *Hsp70* transgene and MCP-GFP were crossed to RNAi flies and dissected and treated as described above. Glands with localized mCherry-LacI fluorescence were imaged using 561 nm (20 µW or 8.5 kW/cm$^2$) and 488 nm (25.25 µW) lasers simultaneously. Nuclei with a visible locus in the 488 channel were scored as being transcribed. Using ImageJ2, the intensity of the locus in the 488 channel was measured and divided by the intensity of a random background region of interest of the same dimensions.

### Reporting summary

Further information on research design is available in the Nature Portfolio Reporting Summary linked to this article.

## Data availability

The data that support this study are available upon request. All sequencing data generated in this study, including processed and normalized files, have been deposited in NCBI GEO under accession number GSE211397. *S. pombe* PRO-seq data was downloaded from GEO accession number GSE102308. Processed/quantified microscopy data, as well source data used for all plots, are provided in the Source Data file. Raw microscopy data is available upon request. Source data are provided with this paper.

## Code availability

Pipelines for data processing as well as all code necessary for analysis and plotting are available on GitHub (https://github.com/mdeber/DeBerardine2023_NatComm_NELF)[97].

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

## Acknowledgements

This research was supported by NIH grants GM025232 and GM139738 to J.T.L., and National Science Foundation Graduate Research Fellowship grant 2139899 to P.P.V. We are grateful to Thomas Graham and Xavier Darzacq (Department of Molecular & Cell Biology, University of California, Berkeley) for valuable discussions and advice concerning the analysis of imaging data, and Warren Zipfel for his help with acquiring imaging data. We thank David S. Gilmour (Department of Biochemistry & Molecular Biology, Pennsylvania State University) for providing fly lines and antibodies. Sequencing was done by Cornell BRC Genomics Core Facility (RRID:SCR_021727). Imaging was done at the Cornell BRC Imaging Core Facility (RRID:SCR_021741) with equipment funded by NYSTEM (C029155) and NIH (S10OD018516).

## Author contributions

G.T.B. and J.T.L. conceived of the project. G.T.B. designed and carried out initial S2 cell experiments. M.D. performed additional experiments and all data analysis. P.P.V. designed, performed, and analyzed imaging experiments, with additional analysis by M.D. Manuscript was written by M.D. with contributions from P.P.V., J.T.L., and G.T.B.

## Competing interests

The authors declare no competing interests.
