## [Peer Review File · Nature Communications]

REVIEWER COMMENTS

Reviewer #1 (Remarks to the Author):

The authors have used RNAi to partially deplete NELFE over 4 days and have analyzed the consequences on Pol II transcription using PRO-seq and RNA-seq. Depletion of NELFE led to reduction of nascent transcripts over pause regions and in gene bodies, but surprisingly almost no effects were observed on the levels of mRNAs. An interesting effect of NELFE depletion was observed when cells were treated with flavopiridol to inhibit P-TEFb during the last few minutes of RNAi treatment. Depletion led to a somewhat subtle increase in nascent transcripts in the first few kb of some gene bodies. The mechanism involved was not elucidated.

Specific Issues:

1. The Western blot analyzing the depletion NELFE in S2 cells (experiments 1 and 2) in Fig. S1 does not have any loading controls and cannot be used to document depletion. Another Western blot analyzing NELFE depletion in the flavopiridol titration experiment in Fig. S4 shows NELFE and alpha Chromator as if the later was used as a loading control. However, the legend states “Alpha-Chromator staining provided for reference, but not relied on as a quantitative loading control.” This leaves the level of depletion undetermined which is critical to evaluate the effects (or lack of effects) seen. Only for the in salivary gland study was a loading control shown (Fig. S6A). Unfortunately it was a subunit of DSIF, the factor that works with NELF to affect pausing. Visual quantification suggests that there was very little depletion of NELF in those experiments.
2. The authors emphasized in the results section that they used mouse spike-ins to normalize the PRO-seq data, but in the Methods section it was stated that the mouse spike-ins were not universally used because of poor reproducibility. The Methods section is confusing because it states that the reads at the ends of long genes was used for normalization and this was applied for the FP conditions for both experiments. If this is the case then I don't understand how the effect of FP can be determined since the non-FP datasets were normalized another way. Exactly what was done needs to be clarified to evaluate the significance of quantitative changes in PRO-Seq signals.
3. Fig. 2 is difficult to follow. Panel A and B are linked by dotted lines, but the legend seems to state that the number of genes in A was 3280 and in B was 469. The number of genes in D was not provided. The heatmaps in E and F are potentially interesting, but only 319 genes were used, not “a large fraction of our gene list” as stated in the Results. Examples should be shown beyond the 2 shown in Fig. 4 in which only 1 of the 2 examples could be seen to exhibit the effect. I took hours to download and examine many of the datasets and found reproducibility to be good (by eye). However, I could not find another gene that exhibited the buildup of Pol II in the early gene body. It would have been nice to have a list of the genes that were determined to display the phenotype. This could have been including

with the PRO-Seq statistics (reads across the genomes utilized, normalization factor calculations etc.) that were also not provided.

4. I could not follow the results in Fig. 3. What is a paused gene? Any gene with a level of paused Pol II that is greater than that found in gene bodies? Any gene with a peak of paused Pol II after flavopiridol treatment? Were absolute levels of paused reads counted or something related to the ratio of paused reads versus gene body reads? It seems to me that it is difficult to make the statements that the subtle bleeding of Pol II into early gene bodies is only present in when flavopiridol is present. It would be much more difficult to determine if there was such a bleeding when normal productive elongation was present (no flavopiridol) because the later could mask the smaller signal. It doesn't really make sense to me that there wouldn't be a signal in the absence of flavopiridol since most polymerases are not acted upon by P-TEFb normally and should behave as if Cdk9 was inhibited.

5. The photoactivation experiments (Fig. 6) did not provide any proof to support the model that Pol II moves into gene bodies in the presence of flavopiridol that distinguishes this study with other published NELF depletion studies. This is due in part to the rather low signal to noise. The authors also seem to be hesitant to draw many conclusions from the experiments which seem to show that there is little transcription in the presence of flavopiridol in NELF depleted glands. This is interpreted to suggest that the Pol II that builds up in the gene body turns over rapidly. Overall, given the complexity of the artificial system, the low signals, the poor depletion of NELF (Fig. S6), and incomplete determination of the position of Pol II in the system, I am not convinced that the results strengthen the other findings.

6. The main issue is that it is difficult to determine how the study significantly impacts our understanding of the function of NELF. The only new finding (which may be specific to *Drosophila*) is that Pol II builds up in early gene bodies in a small fraction of genes when NELF is depleted in the presence of flavopiridol. This was not seen in Aoi, et al. 2020 Mol Cell that performed an extensive analysis of NELF depletion in human cells. That study employed a rapid NELF depletion (2 h) instead of the 4 day depletion utilized here which would minimize secondary effects. Importantly, in neither study is the mechanism of what happens defined. Why does Pol II move further downstream in the absence of NELF? One explanation is that NELF increases pausing (which was documented decades ago) and that loss of the factor frees Pol II to attempt to elongate into or through the first few nucleosomes.

Reviewer #2 (Remarks to the Author):

The manuscript by DeBerardine et al describes a potentially novel aspect of the relationship between two key factors controlling promoter-proximal pol II pausing.

Namely, the Negative ELongation Factor NELF and pause-releasing p-TEFb are more essential for transcription when present together, conceptually resembling a plasmid toxin-antidote relationship. Experimentally, the effects of NELF insufficiency and p-TEFb chemical inhibition were observed in S2 cells separately and together. The bottom line is that lowering NELF levels made paused Pol II proceed further into the gene body when p-TEFb kinase activity was inhibited. It appears as if NELF removal relaxes the requirement for P-TEFb and, therefore, NELF enforces the requirement of metazoan pol II for p-TEFb function, potentially establishing a tighter window for the action of regulatory factors. The manuscript overall makes a number of interesting observations and speculations. The experiment described in Figure 4 regarding the advancing wave in the presence of FP is compelling.

Points.

Figure 1 and the relevant results section. Previous NELF RNAi depletion work in S2 cells showed more extensive perturbation of stable mRNA than what is seen in Fig 1. Could authors comment?

NELF depletion was shown in a couple of Adelman's papers to install nucleosomes instead of Pol II on a number of genes, possibly explaining downregulation of transcription. Doesn't this imply that at least one major effect of NELF depletion is in reduced promoter utilization rather than post-recruitment? Changes in pro-seq signal density at promoters upon NELF depletion are compatible with reduced initiation. Even though the manuscript is focused on what's going on post recruitment, the relative contribution of NELF depletion to both steps should be mentioned.

Regarding the very first Results section's statement that upon NELF depletion gene expression is broadly maintained. This statement is technically correct. However, is this gene expression due to NELF-less transcription or it uses residual NELF? Reduction of gene body signal does not have to be directly proportional to the fold of NELF depletion. This may be mentioned in discussion.

Figure 3A showing that higher promoter density is associated with higher drifting of pol II may be considered trivial. That is, shouldn't more pol II at a promoter result in more pol ii downstream of the same promoter regardless of the reason? The limitations of detection at particular loci would likely be technical. It seems like the entire section of results is going too much into detail leading to a trivial conclusion. My guess is that this section may have been rewritten many times over, but could authors word it better?

Apart from higher steady-state promoter signal, are genes that show the "drifting" of pol II more likely to be activated or repressed upon NELF depletion (no FP), based on pro-seq data?

If reviewer's memory works at all, Lis lab may have previously shown that cells undergo a

quick recovery from P-TEFb inhibition to promptly restore productive transcription. What is going on here: will this advancing wave from Figure 4 eventually become productive or the productive wave will come in later? The longest times here may be overlapping with the onset of recovery shown previously.

Figure 5 conclusion is at this point a speculation. It does make a great story pitch point, but it is not a result. Making this conclusion into a result would require additional arguments such as showing that this transcription in *Drosophila* is functional for mRNA production, which is beyond the scope of the current manuscript. Technically, productive transcription in NELF-depleted cells may well be due to leftover NELF, breaking the analogy with clearly functional transcription in *pombe*. The fold-reduction of transcription, even if NELF dependent, does not have to mirror the fold depletion of NELF so it is hard to deconvolute. It is also unclear what one is supposed to see in Figure 5 exactly. The conclusion is also harder to make due to potential compensation of CDK9 inhibition after a while. Perhaps this point belongs in Discussion with Figure 5 to be dissolved or made part of Figure 7.

I may have missed it, but are shorter versus longer genes affected differently by these perturbations or not, and does this agree with earlier *pombe* analyses that may have been done?

Example genes shown in Figure 4 do not show reduced promoter pol II density upon NELF depletion. Are there genes that correspond to Figure 2A-B metaplots with reduced promoter density and less effect downstream, or this effect of the wave is limited to certain genes that are either not perturbed or activated upon NELF depletion? This may be related to a point above, but with an emphasis on whether the existing metaplot is representative of individual genes' behavior.

Minor points.

The last sentence of the abstract "By introducing a strict checkpoint for Cdk9, the evolution of NELF was likely critical to enable increased regulation of Cdk9 in higher eukaryotes, converting a pre-existing pausing event into a critical point of gene regulation." does not make sense to me even after multiple attempts at it.

Figure 1. Provide a rough estimate for the efficiency of NELF-E depletion?

The title of the first results section mentions pausing being substantially reduced. Does it mean pausing index is reduced?

In authors' best guess, is the ratio of Pol II/NELF or RNA/NELF expected to change upon NELF depletion?

Figure 2D – maybe mark the mentioned second pause zone with an asterisk?

Last sentence of introduction: what exactly is meant by “increased regulation”? It is a commonly overused term that has no meaning – or rather, too many meanings - without additional explanation.

Wording smithing: In the abstract, remove the parentheses leaving the text inside them in. In the abstract: replace “Upon inhibition of Cdk9, cells with NELF efficiently shutdown gene transcription, while defective, non-productive transcription continues unabated in NELF-depleted cells.” With “Upon inhibition of Cdk9, cells with NELF efficiently shut down gene transcription, while in NELF-depleted cells, defective, non-productive transcription continues unabated.” This is to maintain structure and is an optional suggestion.

Introduction: instead of “a potentially prominent rate-limiting step”, should this be “a prominent, potentially rate-limiting step”? Prominence of pausing is not questioned any longer.

“but DSIF also remains” – delete “also”?

Instead of “(as it does in higher eukaryotes)”, find a way to say “unlike”? This reduces ambiguity.

Delete “many” in the last sentence in the introduction. This sentence does not apply to C elegans anyway.

We thank both reviewers for their critical reviews of our manuscript. In response to their feedback, we have added two additional main figure panels (**Figure 2C-D** in the revised manuscript), three additional supplemental figures (**S3E, S4E-F**), and several tables containing sequencing/aligning statistics in the **Supplemental Material & Methods**. This is in addition to various other edits in the main and supplemental texts.

REVIEWER COMMENTS

Reviewer #1 (Remarks to the Author):

The authors have used RNAi to partially deplete NELFE over 4 days and have analyzed the consequences on Pol II transcription using PRO-seq and RNA-seq. Depletion of NELFE led to reduction of nascent transcripts over pause regions and in gene bodies, but surprisingly almost no effects were observed on the levels of mRNAs. An interesting effect of NELFE depletion was observed when cells were treated with flavopiridol to inhibit P-TEFb during the last few minutes of RNAi treatment. Depletion led to a somewhat subtle increase in nascent transcripts in the first few kb of some gene bodies. The mechanism involved was not elucidated.

Specific Issues:

1. The Western blot analyzing the depletion NELFE in S2 cells (experiments 1 and 2) in Fig. S1 does not have any loading controls and cannot be used to document depletion. Another Western blot analyzing NELFE depletion in the flavopiridol titration experiment in Fig. S4 shows NELFE and alpha Chromator as if the later was used as a loading control. However, the legend states “Alpha-Chromator staining provided for reference, but not relied on as a quantitative loading control.” This leaves the level of depletion undetermined which is critical to evaluate the effects (or lack of effects) seen. Only for the in salivary gland study was a loading control shown (Fig. S6A). Unfortunately it was a subunit of DSIF, the factor that works with NELF to affect pausing. Visual quantification suggests that there was very little depletion of NELF in those experiments.

We agree that a loading control would be helpful in S1A. However, we have shown two replicates of NELF depletion using titration series, which provide a rough estimate of knockdown efficiency. From the titration, it is obvious that most NELF-E protein is depleted, although >10% remains. In response to Reviewer #2, we have **added to the figure legend S1A our rough estimate** that 15-30% NELF-E remains.

The second NELF-E knockdown, S5A, does contain a loading control (Chromator) as well as titration series. We initially downplayed the quantitative value of Chromator as a loading control due to some variance between the LacZ replicates when we performed densitometry:

Counting pixel intensity (au) in equally sized boxes across the NELF-E bands (y-axis; 800 nM near-IR fluor.) and the Chromator bands (x-axis, 700 nM), we found some disagreement between the two replicates of LacZ RNAi. This is why we initially de-emphasized the quantitative value of the loading control.

However, densitometry is challenging for various reasons, and looking at the quality of the bands, we would argue that LacZ RNAi replicate 2 (especially the 6-3-1 bands) is the better titration series, and using this we measure 18-20% of NELF-E is remaining. This agrees with our visual estimation from the titration series that clearly <30% of NELF-E remains. If instead we average the two replicates in the densitometry (average the slopes), we find 25% of NELF-E remains. We've **added to figure legend S5A a conservative estimate** that 15-30% of NELF-E remains.

Concerning the NELF-D RNAi Western (S6A), we agree with the reviewer that a DSIF subunit is not an ideal loading control for NELF given its function in gene transcription by Pol II, and we have **clarified this further in the figure legend S6A**. However, we have quantified total protein being loaded and again provided titration series for estimation. Again, the titration series shows that <30% of NELF-D remains (for instance, Luciferase RNAi is nearly beyond detection at 10%, while NELF-D RNAi does so at 30% loading). We have **added this estimate to figure legend S6A**.

We strongly disagree with the assertion that “there was very little depletion of NELF” (in S6A), which is contraindicated not only by the Westerns but by the substantial and obvious phenotypes in the other assays, including in the imaging experiments e.g., Fig6C-E or S6B.

2. The authors emphasized in the results section that they used mouse spike-ins to normalize the PRO-seq data, but in the Methods section it was stated that the mouse spike-ins were not universally used because of poor reproducibility. The Methods section is confusing because it states that the reads at the ends of long genes was used for normalization and this was applied for the FP conditions for both experiments. If this is the case then I don't understand how the effect of FP can be determined since the non-FP datasets were normalized another way. Exactly what was done needs to be clarified to evaluate the significance of quantitative changes in PRO-Seq signals.

While we had higher variance in the FP-treated samples in the first experiment, our second experiment (Fig S5B-C) shows clearly that using long gene end normalization for the FP-treated samples effectively reproduces the normalization factors obtained using the spike-ins, thus demonstrating the validity of this approach for adjusting the first experiment. While we agree the

compound normalization is less elegant, we think our results are quantitatively more accurate (in the first experiment) if we apply this adjustment. However, we have analyzed our data without this adjustment and it does not substantially alter our findings.

Importantly, the use of this compound normalization approach in the second experiment is entirely inconsequential and was only done for the sake of consistency with the first experiment. We would also emphasize that throughout the manuscript, the second experiment clearly validates the findings from the first.

We have added some text to the Methods section to clarify this.

We would also emphasize that essentially all of our results were initially found using only the spike-in normalization before being redone using the more accurate compound normalization approach.

3. Fig. 2 is difficult to follow. Panel A and B are linked by dotted lines, but the legend seems to state that the number of genes in A was 3280 and in B was 469. The number of genes in D was not provided. The heatmaps in E and F are potentially interesting, but only 319 genes were used, not “a large fraction of our gene list” as stated in the Results. Examples should be shown beyond the 2 shown in Fig. 4 in which only 1 of the 2 examples could be seen to exhibit the effect. I took hours to download and examine many of the datasets and found reproducibility to be good (by eye). However, I could not find another gene that exhibited the buildup of Pol II in the early gene body. It would have been nice to have a list of the genes that were determined to display the phenotype. This could have been including with the PRO-Seq statistics (reads across the genomes utilized, normalization factor calculations etc.) that were also not provided.

We thank the reviewer for pointing out how the dashed line connecting 2A and 2B can be interpreted to imply that the underlying data is identical. Our reason for the differing number of genes in the two panels is that we aimed to be inclusive (having more pause regions even if the downstream gene body was filtered due to length or proximity to a downstream PRO-cap site). However, we can appreciate this point and **we’ve added text to clarify the meaning of the dashed line.** In any case, the figure legend is quite explicit and makes clear what is being plotted. We anticipate most readers will appreciate us showing more genes when it is possible to do so.

We have also added text to describe the genes used in the previous Fig2D (now Fig2F in the revised manuscript).

We also appreciate the issue pointed out regarding our claim that “a large fraction of our gene list” show this phenotype, while the heatmap shown only contains a fraction of our gene list. While the reason for this plotting compromise is stated in the legend, we agree that the claim in the text is not directly supported by the heatmap. To address this, **we have added a supplemental plot showing the extent of the phenotype across the gene list (S3E in the revised manuscript).** To summarize the new figure panel, we show (using DESeq2 with apeglm LFC shrinkage estimators*) that 80% of our filtered genes show enriched Pol II density within the early gene body (300-1300 from the TSS) in NELF RNAi +FP vs. LacZ RNAi +FP. [*] *Omitting the LFC shrinkage would obviously increase the percentage, but in a manner that is statistically unfair.*

Regarding the examples shown in Figure 4, cv-c is a gene of rare length in *Drosophila* and the early gene body is not easily seen on the scale shown. The phenotype being shown in Lrch is the same that is quantified across other genes in revised Fig2H (previously 2F), Fig3B-C, FigS4A, Fig4G-H.

However, we appreciate the reviewer’s concern that not enough examples are shown, and we have **added two more example genes to Figure 2.** Looking through many dozens of genes plotted this way, we find that the gene body phenotype described is very well represented across many genes (as revised Fig2H already shows), including numerous cases in which density there increases over time following FP treatment in NELF RNAi cells (as already apparent in Fig4G-H).

We thank the reviewer for pointing out the absence of sequencing statistics and normalization factors. **We have added these tables in the supplemental materials and methods.**

4. I could not follow the results in Fig. 3. What is a paused gene? Any gene with a level of paused Pol II that is greater than that found in gene bodies? Any gene with a peak of paused Pol II after flavopiridol treatment? Were absolute levels of paused reads counted or something related to the ratio of paused reads versus gene body reads? It seems to me that it is difficult to make the statements that the subtle bleeding of Pol II into early gene bodies is only present in when flavopiridol is present. It would be much more difficult to determine if there was such a bleeding when normal productive elongation was present (no flavopiridol) because the later could mask the smaller signal. It doesn’t really make sense to me that there wouldn’t be a signal

in the absence of flavopiridol since most polymerases are not acted upon by P-TEFb normally and should behave as if Cdk9 was inhibited.

We have not defined genes as being paused vs. non-paused. The relationship between this early gene body phenotype and pausing is a numeric relationship with the quantity PRO-seq signal within pause regions under normal conditions, which is shown in Figure 3. We do not use pausing index (ratio of pause to gene body density) directly, but we do better by plotting both the pause and gene body density simultaneously in Figure 3B and Figure S4A.

For clarity, we have to reiterate that we don't see the same pattern of Pol II enrichment within the early gene body under DMSO conditions, see revised Fig2F-H (previously Fig2D-F) or Fig4H. Of course "signal" in the gene bodies of these genes under normal conditions (because they are expressed), but that's not "masking" any enrichment; our PRO-seq has very good depth/coverage for *Drosophila* and is very sensitive, and we would see that enrichment if it were there (as in Fig4H or Fig2). We wonder if the reviewer might be referring to the fact that Pol II density is normally higher in the early gene body region vs. the later gene body. All the same, that is the "normal" state; what's abnormal is that Pol II density remains abnormally present there following FP in NELF-depleted cells.

We've made some **minor edits to the section describing these results** but we struggled to find better ways of emphasizing these points.

We do not agree that "most polymerases are not acted upon by P-TEFb normally". The fact that FP treatment induces such dramatic phenotypes shows that P-TEFb is normally present and acting on Pol II.

5. The photoactivation experiments (Fig. 6) did not provide any proof to support the model that Pol II moves into gene bodies in the presence of flavopiridol that distinguishes this study with other published NELF depletion studies. This is due in part to the rather low signal to noise. The authors also seem to be hesitant to draw many conclusions from the experiments which seem to show that there is little transcription in the presence of flavopiridol in NELF depleted glands. This is interpreted to suggest that the Pol II that builds up in the gene body turns over rapidly. Overall, given the complexity of the artificial system, the low signals, the poor depletion of NELF (Fig. S6), and incomplete determination of the position of Pol II in the system, I am not convinced that the results strengthen the other findings.

While some leaking of Pol II past the canonical pause in NELF-depleted cells has been reported before (for instance, the “second pause” we refer to in the section discussing Figure 1), the response of these cells to Cdk9 inhibition (namely the enrichment of Pol II for several kilobases within the early gene body) is a novel finding.

We disagree that our imaging experiments are limited by “signal-to-noise”. But as we discuss in the text, we do observe variation in the quantity of Pol II imaged at individual loci. However, we show that the increased turnover rate in the NELF RNAi +FP condition is robust to this variation (S6E). We think that a significant portion of this variation is true biological variation: while sequencing experiments combine material from millions of cells, our imaging experiments capture a single locus of a single cell at a given point in time. **We have added a statement to this effect in the results section.**

The reason that we are hesitant to draw many *other* conclusions from live-cell protein kinetics is stated in the text: it is because of the inherent complexity of the underlying processes, and the high amount of inference necessary to state what fractions of Pol II being imaged are involved in what kinds of interactions/activities. We included this statement because, in our opinion, other studies using similar methods frequently impose interpretations that we think are far too detailed and which rely on too many assumptions that cannot be tested.

However, upon review we think that this statement detracts from clarity, and **we have therefore removed it.**

We thank the reviewer for their opinion on the relative value of our live-cell imaging experiments. However, we stand by our argument that the increased rate of turnover of engaged Pol II is a robust result, and that the simplest and most straightforward explanation for this is continued initiation and termination.

6. The main issue is that it is difficult to determine how the study significantly impacts our understanding of the function of NELF. The only new finding (which may be specific to *Drosophila*) is that Pol II builds up in early gene bodies in a small fraction of genes when NELF is depleted in the presence of flavopiridol. This was not seen in Aoi, et al. 2020 *Mol Cell* that performed an extensive analysis of NELF depletion in human cells. That study employed a rapid NELF depletion (2 h) instead of the 4 day depletion utilized here which would minimize secondary effects. Importantly, in neither study is the mechanism of what happens defined. Why does Pol II move further downstream in the absence of NELF? One explanation is that NELF

increases pausing (which was documented decades ago) and that loss of the factor frees Pol II to attempt to elongate into or through the first few nucleosomes.

We agree that a rapid degron approach is better suited to identifying the first-order consequences of a knockdown. However, for reasons we point out in the main text, the previous Aoi et al. study did not adequately address the functional relationship between NELF and P-TEFb, which is of critical importance. We feel that our “steady state” approach of extended NELF depletion in homeostatic cells is well suited to studying how the presence of NELF affects the first-order function of P-TEFb in the minutes following its inhibition, which is the focus of this manuscript.

We think our finding that NELF-depleted cells were unable to efficiently shutdown gene transcription in response to P-TEFb inhibition is a significant addition to the field and will help direct future research regarding the role of Cdk9 in NELF-containing vs. NELF-lacking organisms [like *S. pombe* (Booth et al. 2018)], in particular the new possibilities for extended regulatory functions of P-TEFb in higher eukaryotes like humans.

Reviewer #2 (Remarks to the Author):

The manuscript by DeBerardine et al describes a potentially novel aspect of the relationship between two key factors controlling promoter-proximal pol II pausing. Namely, the Negative ELongation Factor NELF and pause-releasing p-TEFb are more essential for transcription when present together, conceptually resembling a plasmid toxin-antidote relationship. Experimentally, the effects of NELF insufficiency and p-TEFb chemical inhibition were observed in S2 cells separately and together. The bottom line is that lowering NELF levels made paused Pol II proceed further into the gene body when p-TEFb kinase activity was inhibited. It appears as if NELF removal relaxes the requirement for P-TEFb and, therefore, NELF enforces the requirement of metazoan pol II for p-TEFb function, potentially establishing a tighter window for the action of regulatory factors. The manuscript overall makes a number of interesting observations and speculations. The experiment described in Figure 4 regarding the advancing wave in the presence of FP is compelling.

Points.

Figure 1 and the relevant results section. Previous NELF RNAi depletion work in S2 cells showed more extensive perturbation of stable mRNA than what is seen in Fig 1. Could authors comment?

We were also surprised to see that the decrease in PRO-seq signal globally was not recapitulated at the mRNA level. However, we think that the perturbations we achieved with substantial-but-incomplete depletion of NELF are useful for our study. In the text, we point out a possible role for a global feedback stabilization of mRNA (often called mRNA buffering) which is well

documented (at least in yeast) in helping to maintain the global pool of stable mRNA even if active transcription is moderately decreased. We can't comment much beyond our assessment that our S2 cells were able to adjust or compensate for a modest decrease in gene transcription, which also explains why we did not observe any growth defects.

We are not aware of a previous study utilizing spike-in normalization of mRNA levels following NELF depletion, but it's possible that other experiments had potentially higher knockdown efficiencies and more severe consequences, or that their S2 cells (for some reason) were less able to cope with reduced pausing or moderate decreases in global transcriptional output. It's also possible that a different mRNA-seq experiment may have been more sensitive for the purposes of identifying differentially expressed genes.

NELF depletion was shown in a couple of Adelman's papers to install nucleosomes instead of Pol II on a number of genes, possibly explaining downregulation of transcription. Doesn't this imply that at least one major effect of NELF depletion is in reduced promoter utilization rather than post-recruitment? Changes in pro-seq signal density at promoters upon NELF depletion are compatible with reduced initiation. Even though the manuscript is focused on what's going on post recruitment, the relative contribution of NELF depletion to both steps should be mentioned.

We fully agree. We think it entirely possible that some genes may become less active following NELF depletion as a consequence of nucleosomes encroaching on the promoter region following loss of paused Pol II density. What we can say is that, at least for the large majority of genes, we do not see such a severe shutdown of gene transcription that might imply complete inactivation of the promoter. However, this does not rule out in any way the possibility that increased nucleosome density on the promoter might contribute to reduced initiation rates.

We mention this role of pausing in the discussion section, alongside a couple other models for what NELF accomplishes for gene transcription. Ultimately, we have no way of separating these effects from other (potentially secondary) consequences of extended NELF depletion on promoter utilization, and our critical experiments (Cdk9 inhibition) are naturally more focused on post-initiation processes.

Regarding the very first Results section's statement that upon NELF depletion gene expression is broadly maintained. This statement is technically correct. However, is this gene expression due to NELF-less transcription or it uses residual NELF? Reduction of gene body signal does not

have to be directly proportional to the fold of NELF depletion. This may be mentioned in discussion.

We agree that we cannot rule out the importance of residual NELF. Indeed, the change in NELF abundance is much greater than the change in gene body Pol II density. (We have made changes to the legends that document the estimated level of knock down of NELF in our experiments which leaves ~15-30% of normal levels).

Ultimately, a rapid and complete ablation of NELF would allow for different kinds of conclusions to be drawn e.g., the determination of the extent to which pausing is NELF-dependent vs. NELF-independent. As it is, our depletion of NELF is sufficient to produce the critical phenotypes that we describe.

Figure 3A showing that higher promoter density is associated with higher drifting of pol II may be considered trivial. That is, shouldn't more pol II at a promoter result in more pol ii downstream of the same promoter regardless of the reason? The limitations of detection at particular loci would likely be technical. It seems like the entire section of results is going too much into detail leading to a trivial conclusion. My guess is that this section may have been rewritten many times over, but could authors word it better?

We agree that the result of this section is more descriptive/correlative in nature, but all the same, we think it is noteworthy and significant, even if harder to explain.

However, we have to respectfully disagree that the findings in this section are trivial. For one, these observations provide additional supporting evidence for the conclusion that, with NELF depleted, Pol II is able to drift into the gene body in the absence of Cdk9 activity (as we highlight in the text). But aside from this, we're also reporting on a striking phenotype (this accumulation of Pol II density within the first few kilobases of the gene body in the absence of Cdk9), and the findings of this section show a strong relationship of that phenotype to a general feature of genes (pausing intensity). We feel this relationship is really striking and specific (for instance, the DMSO gene body density is not predictive), and that we would be remiss if we did not show this.

Apart from higher steady-state promoter signal, are genes that show the "drifting" of pol II more likely to be activated or repressed upon NELF depletion (no FP), based on pro-seq data?

We thank the reviewer for this interesting suggestion. In general, we have de-emphasized analyzing our NELF depletion data in terms of which genes are more or less sensitive to NELF depletion, for the simple reason that we anticipate secondary consequences of NELF depletion would have accrued over time.

However, we agree with the reviewer that this idea is still worth exploring, and we have **added an additional supplemental figure panel (S4E) to address this**. This panel shows that the gene body response to NELF depletion (DMSO) has little relationship to the quantity of abnormal density observed following FP treatment. **New panel S4F** is also related to this and another comment below.

If reviewer's memory works at all, Lis lab may have previously shown that cells undergo a quick recovery from P-TEFb inhibition to promptly restore productive transcription. What is going on here: will this advancing wave from Figure 4 eventually become productive or the productive wave will come in later? The longest times here may be overlapping with the onset of recovery shown previously.

This is addressed in Figure 5. In *S. pombe*, we previously observed that gene transcription is quickly "restored" following Cdk9 inhibition, but that the new Pol II complexes are moving more slowly than normal. Upon revisiting these results (in Figure 5), we find evidence that those Pol II complexes (in the advancing wave) show the same distinctive (and aberrant) pattern of Pol II density that we see in NELF-depleted *Drosophila* cells (which we detected much more readily because *Drosophila* genes are frequently much longer than *S. pombe* genes). Note as well that in neither case (*S. pombe* or NELF-depleted S2 cells) do Pol II profiles return to normal after Cdk9 inhibition (at least by 20 minutes).

Re-iterating the main text, we conclude that most of those Pol II complexes that advanced into the gene body without Cdk9 activity are not only moving slowly, but they are also incapable of transcribing past the first few kilobases.

Figure 5 conclusion is at this point a speculation. It does make a great story pitch point, but it is not a result. Making this conclusion into a result would require additional arguments such as showing that this transcription in *Drosophila* is functional for mRNA production, which is beyond the scope of the current manuscript. Technically, productive transcription in NELF-depleted cells may well be due to leftover NELF, breaking the analogy with clearly functional transcription in *pombe*. The fold-reduction of transcription, even if NELF dependent, does not

have to mirror the fold depletion of NELF so it is hard to deconvolute. It is also unclear what one is supposed to see in Figure 5 exactly. The conclusion is also harder to make due to potential compensation of CDK9 inhibition after a while. Perhaps this point belongs in Discussion with Figure 5 to be dissolved or made part of Figure 7.

We think these concerns are addressed in other comments (above and below).

I may have missed it, but are shorter versus longer genes affected differently by these perturbations or not, and does this agree with earlier pombe analyses that may have been done?

We only show this for the first *Drosophila* experiment in revised Fig2G-H (previously Fig2E-F). The localization of the FP response phenotype to the early gene body means that there is a relationship to gene length, but the underlying relationship is a consequence of Pol II generally not elongating effectively beyond the first few kb of a gene following Cdk9 inhibition in NELF-depleted cells.

Example genes shown in Figure 4 do not show reduced promoter pol II density upon NELF depletion. Are there genes that correspond to Figure 2A-B metaplots with reduced promoter density and less effect downstream, or this effect of the wave is limited to certain genes that are either not perturbed or activated upon NELF depletion? This may be related to a point above, but with an emphasis on whether the existing metaplot is representative of individual genes' behavior.

We cannot reasonably plot the pause regions on the same scale as the gene bodies. But as we show in Fig1B, NELF depletion reduces Pol II density in virtually every pause region.

Concerning the relationship between the perturbed pausing in NELF RNAi (DMSO) and abnormal Pol II density in the early gene body following FP treatment, we **added a supplemental figure panel (S4F) to address this**. Our finding is that the *change* in pausing following NELF depletion does not correlate to the aberrant gene body density.

Minor points.

The last sentence of the abstract "By introducing a strict checkpoint for Cdk9, the evolution of NELF was likely critical to enable increased regulation of Cdk9 in higher eukaryotes, converting

a pre-existing pausing event into a critical point of gene regulation.” does not make sense to me even after multiple attempts at it.

We have revised this statement to be more direct.

Figure 1. Provide a rough estimate for the efficiency of NELF-E depletion?

We’ve added conservative estimates to S1A, S5A, and S6A figure legends.

The title of the first results section mentions pausing being substantially reduced. Does it mean pausing index is reduced?

It does, although we are more directly referring to changes in Pol II density within pause regions (as in Fig1B), but we also show the comparative changes in the two regions in S2C.

We added text to the legend of S2C to call attention to this.

In authors’ best guess, is the ratio of Pol II/NELF or RNA/NELF expected to change upon NELF depletion?

We have measured the changes (in absolute terms) of nascent RNA and stable mRNA, and from that we would conclude that the changes are not as severe as the changes we estimate for NELFE protein levels.

Figure 2D – maybe mark the mentioned second pause zone with an asterisk?

We have added a mention of the second pause in the figure legend.

Last sentence of introduction: what exactly is meant by “increased regulation”? It is a commonly overused term that has no meaning – or rather, too many meanings - without additional explanation.

We appreciate this feedback and **we have added some additional clarification this in the text.**

Wording smithing: In the abstract, remove the parentheses leaving the text inside them in.
In the abstract: replace “Upon inhibition of Cdk9, cells with NELF efficiently shutdown gene

transcription, while defective, non-productive transcription continues unabated in NELF-depleted cells.” With “Upon inhibition of Cdk9, cells with NELF efficiently shut down gene transcription, while in NELF-depleted cells, defective, non-productive transcription continues unabated.” This is to maintain structure and is an optional suggestion.

We made this change.

Introduction: instead of “a potentially prominent rate-limiting step”, should this be “a prominent, potentially rate-limiting step”? Prominence of pausing is not questioned any longer.

We meant “potentially prominent” in the sense that pausing may be rate-limiting at many genes under many conditions. We agree that this isn’t clear. For the sake of brevity and clarity, **we’ve removed the word “potentially”**.

“but DSIF also remains” – delete “also”?

We’ve rephrased this.

Instead of “(as it does in higher eukaryotes)”, find a way to say “unlike”? This reduces ambiguity.

Done.

Delete “many” in the last sentence in the introduction. This sentence does not apply to C elegans anyway.

Done.

We thank Reviewer 2 for the various and useful suggestions provided. We hope that we were able to address all of their concerns in our revisions.

REVIEWERS' COMMENTS

Reviewer #1 (Remarks to the Author):

The authors have addressed some of my comments and I am not providing more.

Reviewer #2 (Remarks to the Author):

The authors have addressed my concerns satisfactorily overall. Points of uncertainty and interpretation will be hopefully clarified in subsequent studies. I still disagree with the statement in the title of the first Results section as it is vague due to unclear definition of pausing used across the field. This is not a principal point.